# Cadherin clusters stabilized by a combination of specific and nonspecific cis-interactions

**Connor J Thompson[1†], Zhaoqian Su[2†], Vinh H Vu[3], Yinghao Wu[2], Deborah E Leckband[3,4], Daniel K Schwartz[1]***

[1]Department of Chemical and Biological Engineering, University of Colorado Boulder, Boulder, United States; [2]Department of Systems and Computational Biology, Albert Einstein College of Medicine, Bronx, United States; [3]Department of Biochemistry and University of Illinois, Urbana-Champaign, Urbana, United States; [4]Department of Chemical and Biomolecular Engineering, University of Illinois, Urbana-Champaign, Urbana, United States

**Abstract** We demonstrate a combined experimental and computational approach for the quantitative characterization of lateral interactions between membrane-associated proteins. In particular, weak, lateral (cis) interactions between E-cadherin extracellular domains tethered to supported lipid bilayers, were studied using a combination of dynamic single-molecule Förster Resonance Energy Transfer (FRET) and kinetic Monte Carlo (kMC) simulations. Cadherins are intercellular adhesion proteins that assemble into clusters at cell-cell contacts through cis- and trans- (adhesive) interactions. A detailed and quantitative understanding of cis-clustering has been hindered by a lack of experimental approaches capable of detecting and quantifying lateral interactions between proteins on membranes. Here single-molecule intermolecular FRET measurements of wild-type E-cadherin and cis-interaction mutants combined with simulations demonstrate that both nonspecific and specific cis-interactions contribute to lateral clustering on lipid bilayers. Moreover, the intermolecular binding and dissociation rate constants are quantitatively and independently determined, demonstrating an approach that is generalizable for other interacting proteins.

**\*For correspondence:**
Daniel.schwartz@colorado.edu

[†]These authors contributed equally to this work

**Competing interests:** The authors declare that no competing interests exist.

## Introduction

The quantitative characterization of protein interactions on membranes and at buried interfaces, including the measurement of binding constants, is a major challenge due to the limited experimental approaches capable of interrogating molecular interactions in these environments. While it is common to study interactions between extracellular regions of membrane proteins in solution, such experiments are imperfect proxies for measuring actual membrane protein interactions. Apart from the potential impact of domain isolation on protein folding and function, functionally important protein interactions and oligomerization may arise specifically due to constraints imposed by two- or three-dimensional confinement (*Różycki et al., 2010*; *Weikl et al., 2009*). Notably, the immunological synapse is characterized by the spatial and temporal organization of proteins in the gaps between the surface of an antigen presenting cell and a T-cell (*Grakoui et al., 1999*; *Monks et al., 1998*). This organization is attributed in part to the steric segregation of proteins of different sizes and to cytoskeletal interactions (*Qi et al., 2001*; *Schmid et al., 2016*); the understanding of the role of lateral protein interactions in this protein assembly remains incomplete (*Kaitao et al., 2019*). In addition to cadherins, nectins represent another class of membrane proteins whose lateral clusters mediate cell-cell adhesion (*Rikitake et al., 2012*). Distinct lateral (cis) and trans- (adhesive)

interactions between the four members of the nectin family are associated with differentiation and tissue organization. Although it is possible to quantify trans- (adhesive) interactions (*Chesla et al., 1998*; *Chien et al., 2008*; *Wu et al., 2008*), measurements of lateral interactions underlying protein clustering have been inaccessible.

In this context, cadherins pose a particular challenge. Cadherins are transmembrane proteins that mediate cell-to-cell adhesion in all tissues and regulate a range of biological processes, such as tissue rearrangement and formation, cell motility, proliferation, and signaling (*Gumbiner, 1996*; *Gumbiner, 2005*; *Niessen et al., 2011*; *Pla et al., 2001*; *Takeichi, 1995*). Cadherins mediate intercellular adhesion by binding other cadherins on an adjacent cell surface. Notably, cadherins assemble into dense clusters at these adhesive sites, which are important for regulating the permeability of barrier tissues such as the intestinal epithelium (*Brieher et al., 1996*; *Harrison et al., 2011*; *Wu et al., 2015*). The molecular basis underlying cadherin cluster assembly is therefore of great interest because of its importance for tissue functions.

Experimental evidence supports the postulate that cadherin-mediated adhesion and clustering involves both cis- (lateral) and trans-interactions (adhesive) between cadherin molecules on cell surfaces (*Brieher et al., 1996*; *Harrison et al., 2011*; *Wu et al., 2015*). Early comparisons of cadherin extracellular domain adhesive activity suggested that the protein functions as a cis-dimer, and crystal structures suggested a plausible cis-binding interface (*Brieher et al., 1996*; *Harrison et al., 2011*). Moreover, mutating one or two key amino acids in the postulated cadherin cis-binding interface results in impaired intercellular adhesion and reduced cadherin clustering at cell-cell contacts (*Erami et al., 2015*; *Harrison et al., 2011*; *Shashikanth et al., 2016*; *Wu et al., 2015*). However, despite experimental evidence for the importance of cis-interactions in cell adhesion, they have been difficult to investigate directly (*Brieher et al., 1996*; *du Roure et al., 2006*; *Harrison et al., 2011*; *Hong et al., 2013*; *Indra et al., 2018*; *Klingelhöfer et al., 2002*; *Leckband and Sivasankar, 2012*; *Leckband and de Rooij, 2014*; *Shapiro et al., 1995*; *Troyanovsky et al., 2015*; *Troyanovsky et al., 2007*; *Troyanovsky et al., 2003*; *Wu et al., 2015*; *Yap et al., 1997*; *Yap et al., 1998*; *Zhu et al., 2003*). Due to the relatively weak nature of cis-interactions, traditional solution-phase studies have failed to detect them, even at high protein concentrations (*Häussinger et al., 2004*; *Koch et al., 1999*). Furthermore, attempts to stabilize weak cis-interactions through chemical crosslinking in solution were unsuccessful (*Zhang et al., 2009*).

Computational models of cadherin binding subsequently suggested that the reduction of configurational and orientational entropy under two- and three-dimensional confinement could potentiate cis-interactions. Specifically, the models predicted that tethering cadherin extracellular domains to a two-dimensional (2D) surface, such as a supported lipid bilayer or cell membrane would increase the effective binding affinities of both cis-and trans-interactions (*Harrison et al., 2011*; *Wu et al., 2010*; *Wu et al., 2011*). Unfortunately, measurements based on analyses of photon counting histograms were unable to detect cis-interactions between E-cad extracellular domains on supported bilayers independent of trans-interactions, likely due to the modest cadherin surface concentrations studied (*Biswas et al., 2015*). However, the prediction that membrane-tethered cadherins can form clusters under 2D confinement was recently confirmed indirectly via single-molecule tracking, based on measurements of the diffusion of E-cadherin extracellular domains on supported lipid bilayers, over a very large range of cadherin surface coverage (*Thompson et al., 2019*). Comparisons of wild-type and cis-mutants confirmed that a specific cis-binding interface mediated clustering in the absence of *trans* interactions. Importantly, the diffusion coefficient served as a very sensitive proxy for cis-interactions, because clusters diffuse more slowly than monomers. These findings suggested that cis-interactions between E-cad extracellular domains can result in the formation of large clusters, in the absence of trans-interactions, for cadherin surface coverage above a threshold of ~1,100 E-cad/$\mu$m$^2$(*Thompson et al., 2019*). However, a quantitative understanding of cis-interaction contributions to the assembly of adhesive junctions has been hindered by the lack of approaches capable of identifying and quantifying relevant binding interactions.

Here we used intermolecular single-molecule Förster Resonance Energy Transfer (FRET) microscopy to characterize the dynamic interactions between E-cad extracellular domains tethered to mobile supported lipid bilayers, while simultaneously tracking the motion of E-cad monomers and clusters to determine their diffusion coefficients, and thereby infer their hydrodynamic diameters. By comparing the behavior of wild-type E-cad to that of a mutant that is incapable of specific cis-interactions, we identified two distinct types of lateral interactions, which we attributed to nonspecific

(i.e. not through the specific cis-interface observed in the crystal structure) interactions (present for both wild-type and mutant E-cad) and specific interactions (present only for wild-type E-cad). The specific interactions were significantly stronger, resulting in longer intermolecular associations and a steady-state cluster distribution with a larger characteristic cluster size. Complementary off-lattice kinetic Monte Carlo simulations were performed under conditions designed to mimic the experiments. The kinetic parameters associated with the simulations were constrained by experimental values when applicable; the remaining parameters were optimized so that the steady state cluster size distributions matched those observed experimentally. The experiments and simulations were internally consistent, with a single set of parameters for all experimental conditions. These simulation results suggested that the dissociation rate for specific cis-interactions was approximately 10x slower than for nonspecific interactions under the conditions of the experiments. Thus, while associations due to nonspecific interactions were significantly weaker than cis-interactions, they were substantial and could not be ignored. The simulations also suggested that associations due to cis-interactions were more efficient and likely to occur, than nonspecific interactions. Importantly, the methods developed and employed here can be generally applied to study the dynamics of specific and nonspecific lateral interactions between a wide range of membrane proteins.

## Results

### Nonspecific and specific cis-Interactions are present in E-cad clusters

In order to study E-cad lateral interactions under 2D confinement, donor (Alexa 555) labeled, acceptor (Alexa 647) labeled, and unlabeled E-cad extracellular domains were simultaneously bound to a supported lipid bilayer via hexahistidine-NTA associations and imaged using a prism-based total internal reflection fluorescence (TIRF) microscope. This allowed the observation of a large number of single molecule trajectories at high or intermediate protein surface coverage. Two discrete populations were observed corresponding to negligible energy transfer (low-FRET) and complete energy transfer (high-FRET) (*Figure 1A* and *Figure 1—figure supplement 1*). *Figure 1A* shows a representative FRET heat map showing two distinct populations at high and low FRET efficiency. Each molecular observation within each trajectory was then classified as either a high-FRET or low-FRET efficiency state (where high-FRET corresponds to a putative cis-association) based on the donor and acceptor intensities using an algorithm described previously, allowing the identification of high-FRET and low-FRET time intervals (*Figure 1A* and *Figure 1—figure supplement 1*; *Chaparro Sosa et al., 2018*). Previously, the high-FRET state has been shown to indicate binding (*Kastantin et al., 2017*; *Langdon et al., 2015*; *Langdon et al., 2014*; *Monserud et al., 2016*; *Monserud and Schwartz, 2016*; *Traeger et al., 2019*; *Traeger and Schwartz, 2017*; *Traeger and Schwartz, 2020*). In order to distinguish the effects of specific cis-interactions, E-cad extracellular domain constructs of wild-type E-cad and the cis-binding mutant L175D were used in separate experiments; this particular point mutant was previously shown to be incapable of interacting through the cis-interface (*Harrison et al., 2011*; *Thompson et al., 2019*). Therefore, at similar surface coverage, any difference in apparent interactions between the wild-type and this mutant should primarily be due to the presence or absence of specific cis-interactions.

Three conditions were studied: high-coverage wild-type (~1,400 E-cad/$\mu m^2$), high-coverage mutant (~1,300 E-cad/$\mu m^2$), and intermediate-coverage wild-type (~1,000 E-cad/$\mu m^2$), where these coverage values were chosen based on previous experiments, which demonstrated the onset of significant clustering at surface coverages above ~1,100 E-cad/$\mu m^2$ (*Thompson et al., 2019*). Quantifying cis-interactions independent of trans-interactions at these high surface coverage values is directly physiologically relevant, as cell-cell junctions consist of both adhesive and nonadhesive clusters, and can reach a maximum local surface coverage of ~49,000 E-cad/$\mu m^2$ (*Indra et al., 2018*; *Wu et al., 2015*). A total of ~4000 trajectories were observed, of which ~750 exhibited FRET events, consisting of ~85,000 total molecular observations at each of the three experimental conditions employed. *Supplementary file 1a* contains the exact number of total trajectories, trajectories exhibiting FRET association, and total number of displacements for each experimental condition. To permit single molecule localization, the donor-labeled E-cad concentration was kept very low as described in the Materials and methods section. The acceptor-labeled E-cad concentration was much larger than that of the donor, allowing the observation of a large number of FRET events and

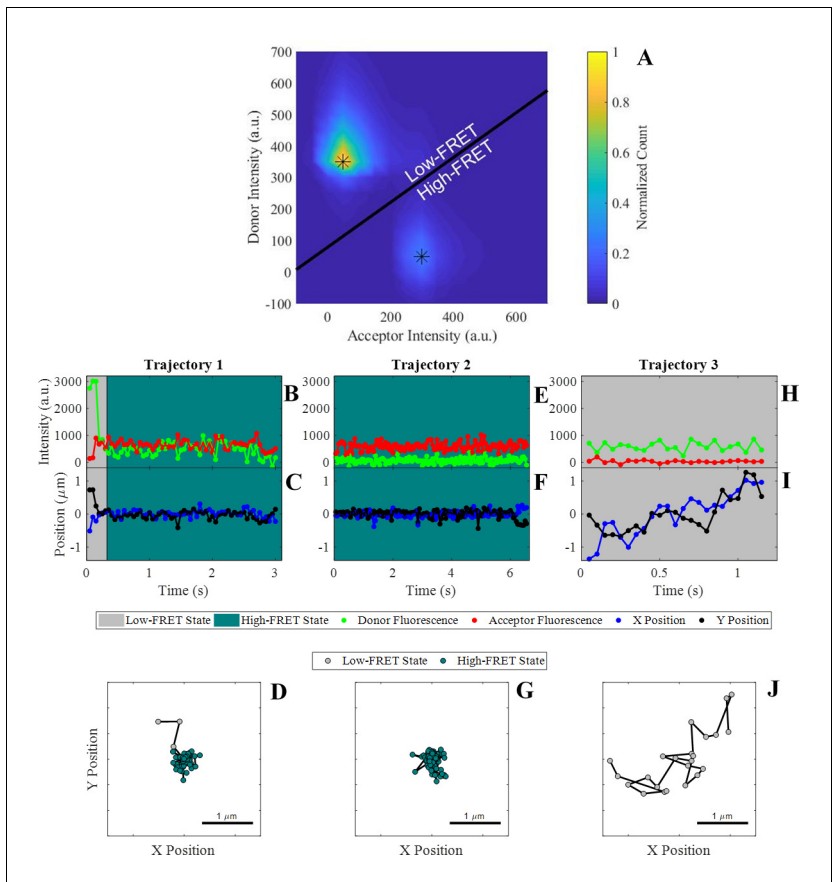

**Figure 1.** Observation of cis-interactions via single-molecule FRET. (**A**) Representative heat map of donor and acceptor intensities showing two populations at high and low FRET efficiency indicated by the asterisks. The black line represents the threshold between the two states used to assign each observation to the high or low-FRET state. (**B, E, H**) Donor and acceptor trajectories for a FRET pair throughout representative trajectories, which are used to determine if the donor E-cad molecule is in a high-FRET or low-FRET state. (**C, F, I**) X and Y Cartesian coordinates for the donor or acceptor molecule over the length of the trajectory. (**D, G, J**) Two dimensional trajectory plots of the same trajectories, where the symbol color corresponds to the assigned FRET-state. The background of the trajectory time traces for intensity and position indicate the assigned FRET-state.

The online version of this article includes the following figure supplement(s) for figure 1:

**Figure supplement 1.** Heat maps showing binned acceptor and donor intensities using all molecular observations.

**Figure supplement 2.** Overall complementary cumulative distributions of squared displacement calculated using only trajectories from a single movie for all three experimental conditions.

**Figure supplement 3.** Complementary cumulative association time (high-FRET state dwell time) distributions calculated using only trajectories from a single movie for each of the three experimental conditions.

**Figure supplement 4.** Fluorescence recovery after photobleaching (FRAP) analysis indicates the formation of a continuous, fluid supported lipid bilayer.

**Figure supplement 5.** All localized and tracked trajectories two frames and longer from the 30 s high surface coverage wild-type movie clip included as *Video 1* (left).

**Figure supplement 6.** Representative histogram of positional localization uncertainties for all molecular observations for the high surface coverage wild-type condition indicating a significant number of observations with a large position uncertainty comparable to the size of a pixel (0.43 µm).

**Figure supplement 7.** Trajectory median donor intensity histograms showing the 60th percentile cutoff as a vertical red line used to remove bright contaminants, donor E-cad aggregates, and donor E-cad labeled with multiple fluorophores.

ensuring that multiple acceptors were present in clusters. Due to limitations in acceptor

concentration caused by the need to avoid excess background that results from direct excitation of the acceptor, unlabeled E-cad was added to reach sufficiently high surface coverages required for cluster formation.

In some trajectories, transitions between FRET states were observed, presumably indicating association and dissociation events between donor and acceptor labeled E-cad. However, many trajectories showed no FRET-state transitions, where a trajectory began by either adsorption or diffusion into the field of view in a given state and remained in that state until the trajectory ended through desorption, diffusion out of the frame, or photobleaching. Representative trajectories illustrating these different situations are shown in *Figure 1B–J*. For example, in trajectory one, the donor E-cad begins in the low-FRET state and appears to be diffusing quickly, based upon the large positional fluctuations. After ~0.33 s, a transition from low to high FRET-state indicates the association of the donor E-cad with a cluster. This FRET transition coincides with a significant decrease in the positional fluctuations, consistent with the motion of a large cluster. In contrast, representative trajectory two exhibits no apparent FRET-state transitions. The trajectory begins in the high-FRET state and remains in this state throughout the entire trajectory. The position fluctuations are small, and the molecule remains in a small, confined, region. This behavior suggests that the donor E-cad is associated with a large cluster that contains one or more acceptor E-cad molecules. Lastly, trajectory three remains in the low-FRET state throughout the entire trajectory, and exhibits large positional fluctuations, consistent within an unassociated monomer of donor E-cad.

As is apparent from *Figure 1B–J*, transport properties are often coupled to the FRET-state of a molecule. This is because the FRET-state reflects the oligomeric state of an E-cad molecule, and large oligomers diffuse slower than a monomer due to increased protein-lipid interactions, which is the primary source of drag (*Cai et al., 2016*). In order to assess this hypothesis and confirm that the high-FRET state does in fact correlate with protein clusters involving a donor and one or more acceptors, the average short-time diffusion coefficient ($\overline{D}_{short}$) was determined for the high and low FRET-state populations independently. This was done by constructing complementary cumulative squared displacement distributions (CCSDDs) for each state, under each experimental condition, and then fitting these distributions to a Gaussian mixture model containing three terms (See Materials and methods section for more details on distribution calculations, fitting, and $\overline{D}_{short}$ calculation). $\overline{D}_{short}$ represents the average instantaneous molecular diffusion coefficient at the shortest experimentally accessible time scale and is especially useful for systems where molecules change FRET states within a trajectory (*Chaparro Sosa et al., 2020*; *Chaparro Sosa et al., 2018*; *Langdon et al., 2015*). Additionally, overall CCSDDs were constructed, in order to determine overall values of $\overline{D}_{short}$ under each experimental condition. Overall CCSDDs and Gaussian mixture model fits are shown in *Figure 2—figure supplement 1*. *Figure 2A–C* shows the CCSDDs for both FRET-states (at each of the three experimental conditions) with the respective Gaussian mixture model fits. The FRET-state CCSDDs (*Figure 2A–C*) indicate that the probability of a large displacement is significantly smaller for E-cad in the high-FRET state for all conditions. *Figure 2D* shows the resulting values of $\overline{D}_{short}$ determined from the fit parameters. *Supplementary file 1b* shows all CCSDD fit parameters.

Most importantly, *Figure 2D* shows that the values of $\overline{D}_{short}$ are significantly smaller for the high-FRET state relative to the low-FRET state. This behavior is consistent with the interpretation that the high-FRET state corresponds to E-cad in an associated state, where it diffuses as an oligomer or large cluster. Of course, due to the presence of unlabeled E-cad, it is possible for an E-cad donor molecule to be associated with a cluster but remain in a low-FRET state. The low-FRET state population comprises a combination of unassociated donor E-cad and donor E-cad that is associated with unlabeled E-cad; consequently, this population is more complicated to interpret. Nevertheless, the inclusion of monomers in the low-FRET state (and not the high-FRET state) is expected to result in larger values of $\overline{D}_{short}$ for the low-FRET state, as observed for all three experimental conditions, even for the mutant that cannot interact through the cis-interface. Importantly, the observation that the mutant also exhibits decreased diffusion in a high-FRET state (from $0.569 \pm 0.008\ \mu\mathrm{m}^2/\mathrm{s}$ to $0.44 \pm 0.01\ \mu\mathrm{m}^2/\mathrm{s}$) suggests that the proteins can associate by nonspecific interactions in addition to the specific cis-binding interface expected for wild-type E-cad.

As shown in *Figure 2D*, the average protein diffusion associated with both of the FRET states is slowest at the higher surface coverage of wild-type E-cad. This observation is consistent with the presence of more large protein clusters than at lower surface concentrations or in the absence of

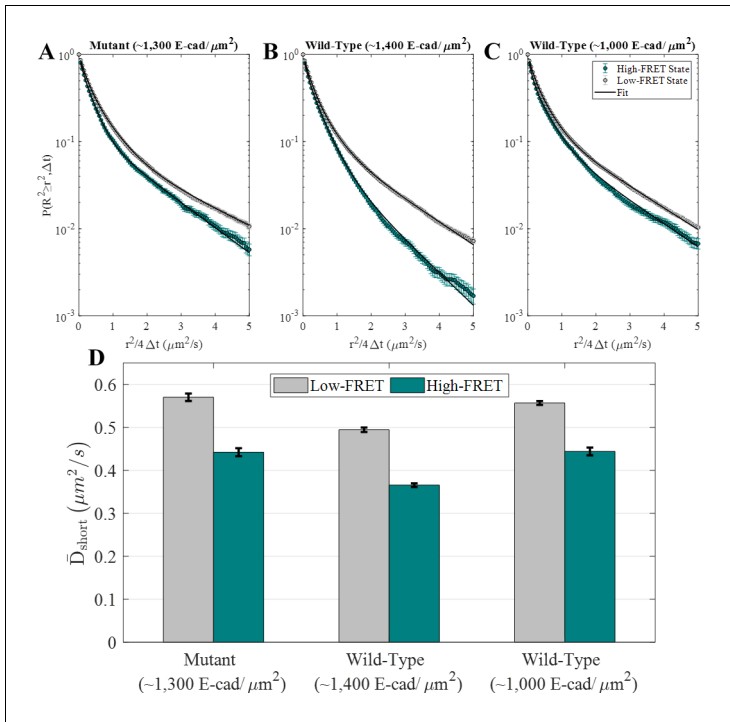

**Figure 2.** E-cad diffusion depends on FRET state and interaction capability. (**A–C**) Complementary cumulative squared displacement distributions in the high-FRET and low-FRET states for the mutant and two wild-type E-cad conditions, along with the respective Gaussian mixture model fits. Error bars correspond to the standard deviation of CCSDDs calculated using 100 samples using a bootstrap method with replacement and are generally smaller than the data points, except in the 'tail' of the high-FRET state distributions. (**D**) $\overline{D}_{short}$ in the high-FRET and low-FRET states for the mutant and two wild-type conditions. Error bars represent the standard deviation of fitting 100 samples using a bootstrap method with replacement.

The online version of this article includes the following figure supplement(s) for figure 2:

**Figure supplement 1.** Overall CCSDDs, using all displacements from both high and low-FRET states, for the mutant and two wild-type E-cad conditions.

**Figure supplement 2.** Displacement-based trajectory filtering results in short-time diffusion expected for supported lipid bilayers.

---

specific cis-interactions. The formation of these large clusters is presumably supported by a large number of nonspecific interactions, in combination with frequent cis-interactions at the higher concentration. Interestingly, wild-type E-cad at lower surface concentration and mutant E-cad at higher concentration exhibit similar diffusion constants for both FRET-state populations, suggesting that the average cluster sizes are comparable in these two systems, due to a balance between the strength and frequency of nonspecific and specific interactions. This is consistent with previous findings that specific cis-interactions between wild-type E-cad proteins primarily affected diffusion only at surface coverages above ~1,100 E-cad/µm² , while nonspecific interactions between mutant E-cad did not cause significant slowing even above this threshold (*Thompson et al., 2019*). Additionally, the overall $\overline{D}_{short}$ values (*Supplementary file 1b*) show that effective total diffusion was slowest for high surface coverage wild-type E-cad, and that the overall diffusion for mutant E-cad and intermediate coverage wild-type E-cad was similar. To better understand the relationship of nonspecific and specific lateral interactions between E-cad extracellular domains, a detailed investigation of interaction dissociation kinetics was performed as described below.

## Nonspecific Cis-Interactions dissociate faster than specific Cis-Interactions

Classifying each observed trajectory into the high-FRET or low-FRET state provides information about the time intervals spent in each state (dwell time), in addition to the state-dependent

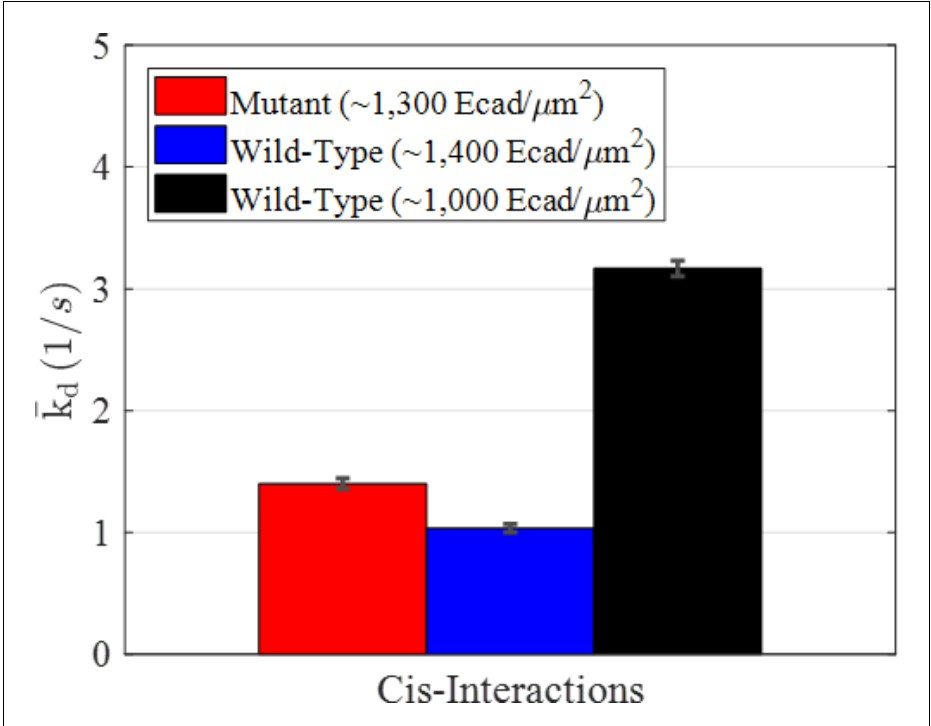

**Figure 3.** Average dissociation rate constants ($\bar{k}_d$) for the mutant and two wild-type conditions resulting from modeling interactions using a Markov model. Error bars were estimated as the square root of the Cramèr-Rao lower bound.

The online version of this article includes the following figure supplement(s) for figure 3:

**Figure supplement 1.** Complementary cumulative association time (high-FRET state dwell time) distributions calculated for each of the three high surface coverage experimental conditions and the low coverage control.

**Figure supplement 2.** High-FRET and low-FRET complementary cumulative surface residence time (observation time) distributions for mutant E-cad and two concentrations of wild-type E-cad.

**Figure supplement 3.** Complementary cumulative state dwell time distributions for the high and low-FRET states for the mutant and two wild-type E-cad conditions, compared to the predicted state dwell time distributions based upon the three-state, heterogeneous Markov model maximum likelihood estimate with beta-distributed transition probabilities.

**Figure supplement 4.** Probability density functions for the state transition rates between the high and low-FRET states for the mutant and wild-type conditions determined based upon the Markov model estimated, beta-distributed transition probabilities.

**Figure supplement 5.** Low-FRET state complementary cumulative dwell time distributions for the mutant and two wild-type conditions.

**Figure supplement 6.** Beta distributions of state transition probabilities between the high and low-FRET states for the mutant and two wild-type conditions corresponding to the Markov model maximum likelihood estimated beta distribution parameters.

transport properties discussed previously. The dwell times in each state contain direct information about the nature and energies of interactions. These data can be used in tandem with the transport information, which provides indirect information about clustering. High-FRET state dwell time distributions are shown as *Figure 3—figure supplement 1* and generally indicate longer dwell times for wild-type compared to mutant E-cad, and that dwell time generally increases with surface coverage. In particular, inspection of the dwell time distributions (*Figure 3—figure supplement 1*), in conjunction with the high-FRET surface residence time distributions (*Figure 3—figure supplement 2*), suggests that the higher probability of long dwell times for wild-type E-cad are due to stronger interactions. However, it is challenging to extract quantitative information directly from the dwell time distributions for a number of reasons, such as: heterogeneity in the number of fluorescent labels per E-cad, differences in labeling efficiency between the wild-type and mutant, and the

convolution of photobleaching and desorption with the dwell times. Therefore, in order to rigorously extract quantitative dissociation rates, it was advantageous to employ a three-state Markov model that accounted for trajectory observation times, as described in detail below.

A three-state Markov model that has previously been used to model protein conformations based on intramolecular FRET time series data (*Kienle et al., 2018*) was used to quantitatively model intermolecular FRET time series data associated with E-cad interactions in this system. This model incorporated three states: high-FRET, low-FRET, and off, where the off-state corresponded to the end of a trajectory due to photobleaching, desorption from the surface, or diffusion out of the field of view. To account for heterogeneity in protein interactions, a beta distribution of state transition probabilities between the high-FRET and low-FRET states was incorporated into the model. This heterogeneity reflects the diversity of local environments, including various cluster sizes, shapes, etc. A maximum likelihood estimate of the beta distribution parameters was iteratively generated based on the previously assigned sequence of states for each trajectory, and the average interaction rates for transitions from the low-FRET state to the high-FRET state and vice versa were determined. Here, the average interaction rate for transition from the high-FRET state to the low-FRET state was equivalent to the average dissociation rate constant ($\bar{k}_d$) for this system due to the concentration independence of the dissociation reaction rate. For additional details of the model, see the Materials and methods section and the previous application of this model to protein conformational changes (*Kienle et al., 2018*). To confirm the accuracy of modeling the observed interactions, complementary cumulative dwell time distributions were generated for comparison with measured distributions, by using the maximum likelihood estimated transition probabilities; they are presented as *Figure 3— figure supplement 3*.

As shown in *Figure 3*, $\bar{k}_d$ varied significantly between wild-type and mutant E-cad, and also between wild-type E-cad at high and intermediate surface coverage. The values of $\bar{k}_d$ were $1.40 \pm 0.04\ s^{-1}$, $1.04 \pm 0.03\ s^{-1}$, and $3.17 \pm 0.06\ s^{-1}$ for the mutant, at high wild-type surface coverage, and at intermediate wild-type surface coverage, respectively. Thus, wild-type E-cad at high surface coverage exhibited the slowest dissociation (i.e., the most stable clusters), consistent with expectations from the FRET-state diffusion analysis. This is plausible, since larger clusters at higher surface concentrations were expected to enable both long-lived multivalent nonspecific interactions as well as a significant number of longer-lasting specific cis-interactions. For mutant E-cad at high coverage, the value of $\bar{k}_d$ was larger than for wild-type E-cad at high surface coverage, but significantly smaller than for wild-type E-cad at intermediate surface coverage. This was presumably due to the relatively high effective strength of nonspecific interactions, at high surface coverage, due to avidity and trapping effects. Finally, the largest value of $\bar{k}_d$ (i.e., the least stable clusters) was observed for wild-type E-cad at intermediate surface coverage, due mainly to the frequent and short-lived nonspecific interactions. This is consistent with previous observations and suggests that specific cis-interactions were infrequent at this intermediate surface coverage.

Overall, an additional interesting result from the modeling of the FRET time-series data was that E-cad interactions were highly heterogeneous under all conditions, as indicated by the distributions of dissociation rates (*Figure 3—figure supplement 4*), presumably due to the wide variety of cluster sizes and shape, the presence of trapping and avidity effects, and the complex combination of specific and nonspecific interactions. The mutant E-cad interactions, which included only nonspecific associations, were also heterogeneous; perhaps reflecting the potential for multivalency in these associations. Nonspecific interactions also appear to be surface coverage dependent, suggesting increasing effective strength with increasing surface coverage likely due to binding avidity within large protein clusters and the prevalence of steric effects such as trapping within cluster interiors, consistent with previous observations (*Langdon et al., 2014*). Moreover, the presence of both nonspecific and specific interactions creates many complex scenarios, including the potential for specific cis-interactions to form via an initial nonspecific 'encounter complex' that transitions to the specific cis-interaction through orientational changes. To capture this complexity directly, explicit kinetic Monte Carlo simulations were performed, as described below.

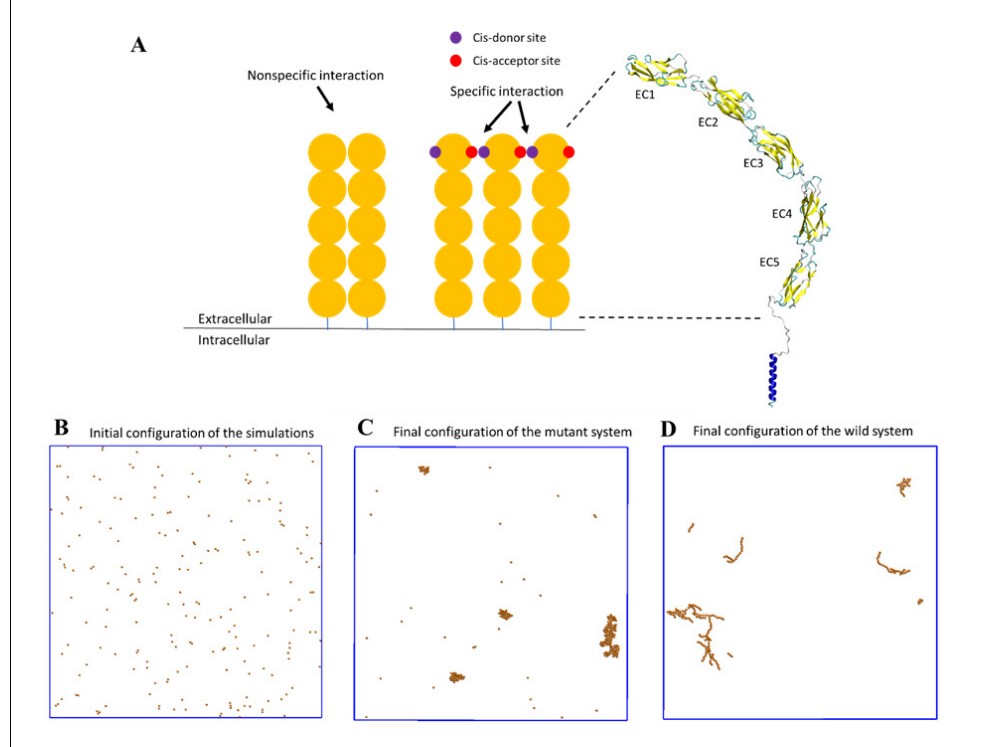

**Figure 4.** A coarse-grained model was constructed to simulate the spatial-temporal process of E-cad clustering. (A) E-cad extracellular domains (orange), nonspecific and specific cis-interactions. Cis-donor sites are labeled in purple, and cis-acceptor sites are labeled in red. A structural model of the E-cad is shown on the right side. Ectodomain structure with EC domains 1–5 numbered from the N-terminus. (B) Top view of initial configuration in the simulations. The number of E-cad molecules is equal to 200. (C) Top view of final configuration in the mutant system. (D) Top view of final configuration in the wild-type system.

## Heterogeneous kMC simulations differentiate specific and nonspecific interactions

The single molecule FRET results provided novel insights into the qualitative overall behavior of lateral interactions between E-cad extracellular domains tethered to a supported bilayer. They also enabled quantitative characterization of the dissociation kinetics due to specific and/or nonspecific interactions. Nevertheless, gaps remained in the understanding of the physical basis of the observations. In particular, as discussed above, it was difficult to unambiguously distinguish association events. Additionally, single molecule FRET permitted the assignment of only two states: low-FRET and high-FRET (associated). Therefore, for a system in which intrinsically different (and highly heterogeneous) interactions are expected, these experimental observations could not distinguish between the different types of interactions underlying clustering. Nor could we quantitatively extract the independent contributions and kinetics of each interaction. To address these experimental limitations, kinetic Monte Carlo (kMC) simulations were performed. Importantly, these simulations incorporated both the nonspecific and specific interactions revealed by the FRET data.

To model specific interactions, each wild-type E-cad molecule had one cis-donor site and one cis-acceptor site located on opposing sides of the molecule (see *Figure 4A*), in order to incorporate the specific orientational constraint associated with specific cadherin cis-interactions (*Harrison et al., 2011*). This allowed each E-cad molecule to participate in a maximum of two specific cis-interactions and mandated the formation of flexible linear oligomers. The inclusion of nonspecific interactions was then accomplished by allowing additional interactions in all directions, within a specified distance constraint. By allowing molecules to form both nonspecific and specific interactions, association and dissociation rate constants could be tuned independently for both interactions.

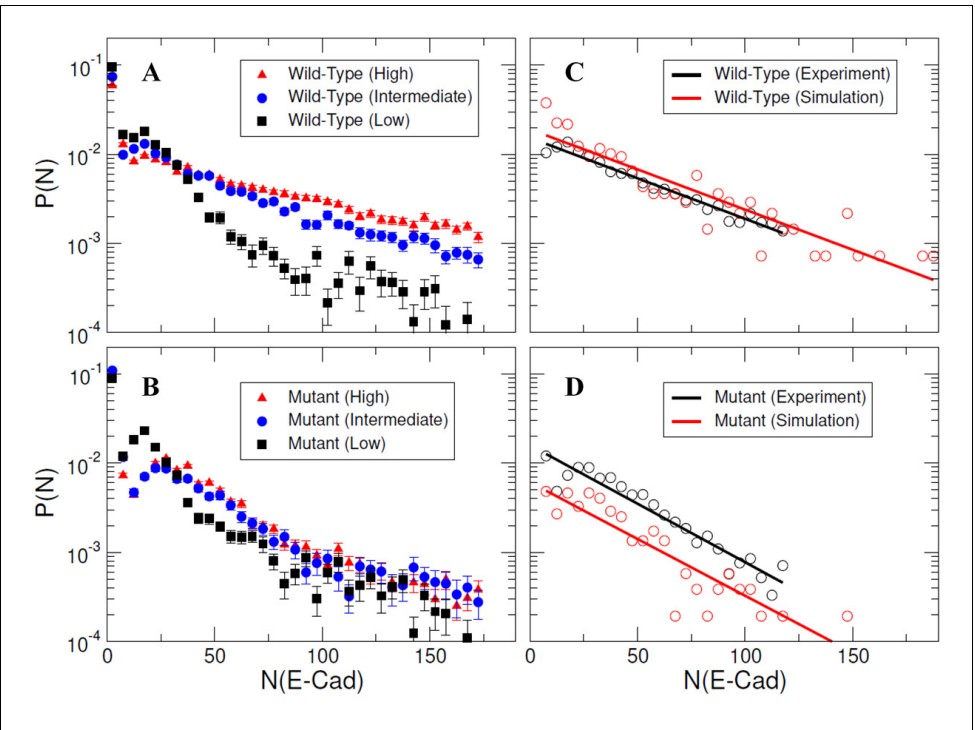

**Figure 5.** Specific and nonspecific interactions can cause E-cad clustering. (**A–B**) Representative experimental cluster size probability distribution functions for wild-type and mutant E-cad at low, intermediate, and high surface coverages. Error bars correspond to the standard deviation of cluster size probability distribution functions calculated using 100 samples using a bootstrap method with replacement. (**C–D**) The comparison of experimental and simulated cluster size distributions for mutant and wild-type E-cad. The solid lines indicate the single exponential fitting.

The online version of this article includes the following figure supplement(s) for figure 5:

**Figure supplement 1.** Distributions of mutant E-cad cluster size for different combinations of nonspecific interaction on/off rate.

**Figure supplement 2.** Distributions of wild-type E-cad cluster size for different combinations of specific interaction on/off rate.

**Figure supplement 3.** E-cad is primarily bound to a single lipid.

**Figure supplement 4.** Trajectory averaged friction factor probability distributions for wild-type (top) and mutant (bottom) E-cad at high, intermediate, and low E-cad surface coverage.

**Figure supplement 5.** Distribution of cluster size for wild-type and mutant E-cad at a surface density of 312.5 E-cad/μm², 625 E-cad/μm², and 1,250 E-cad/μm², respectively.

We computationally simulated the clustering of E-cad on supported lipid-bilayers, using a domain-based, coarse-grained model (*Figure 4A*). After random initial placement, all molecules and clusters stochastically diffused off-lattice, using periodic boundary conditions. The average cluster size was monitored throughout the simulation period. Simulations were run until the average cluster size did not change significantly. This implied that equilibrium was reached, analogous to the experiments. A total of 50 simulations were run at three different surface coverages (312.5 E-cad/μm², 625 E-cad/μm², and 1,250 E-cad/μm²) for both wild-type and mutant E-cad. Simulations with wild-type E-cad included both nonspecific and specific interactions, but simulations of cis-mutants allowed the proteins to associate only by nonspecific interactions. Simulations also used different combinations of binding rates within a biologically relevant range. For additional details on kMC simulations, see the Materials and methods section.

For each set of simulation parameters, multiple independent trajectories were generated to assure that the computational data were statistically meaningful. Detailed strategies of the sensitivity analysis are summarized in the Materials and methods section. At the end of the simulations, the cluster size distributions were calculated by averaging from all the trajectories in the systems. In

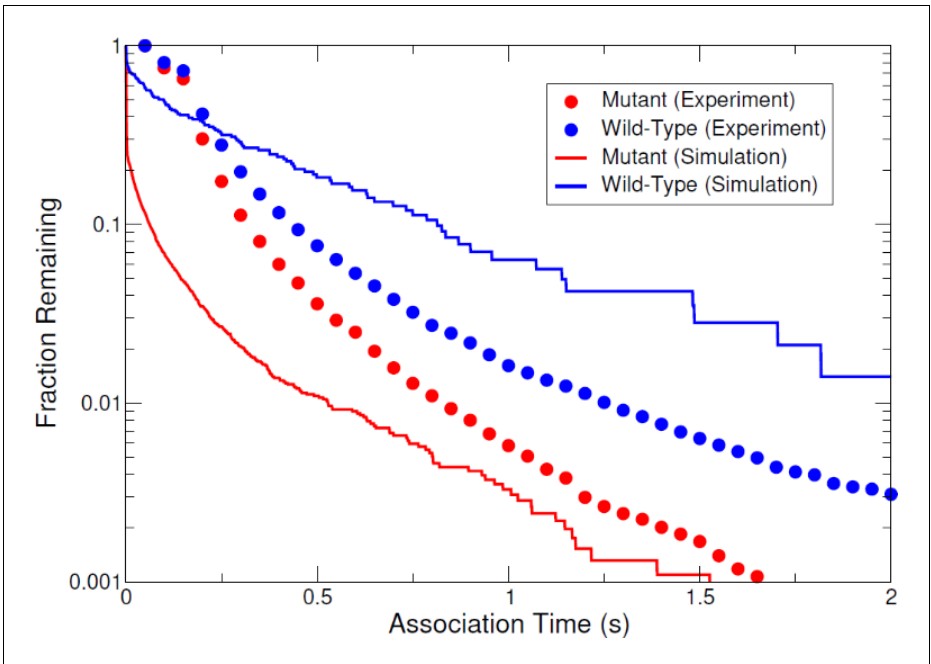

**Figure 6.** The comparison of experimental and simulated complementary cumulative association time distributions for mutant and wild-type E-cad.

order to directly compare the cluster size distributions from simulations with the experimental distributions, similar surface coverages were considered between the simulation and experimental systems.

To allow direct comparison of kMC simulations to experimental results, E-cad cluster size probability distributions were calculated using raw trajectory friction factor data adapted from *Thompson et al., 2019*, as described in the Materials and methods section. Resulting experimental cluster size probability distributions are shown as *Figure 5A–B*, for both wild-type and mutant E-cad at high, intermediate, and low surface coverages corresponding to ~39,000 E-cad/µm², ~1,000 E-cad/µm², and ~0.6 E-cad/µm², respectively. However, due to the dynamic nature of cis-interactions and the trajectory filtering method, the relative change in cluster size distributions with coverage and between wild-type and mutant is most relevant. For mutant E-cad, the change in the cluster size distribution with increasing surface coverage is subtle, and mainly visible in the small cluster regime, where the peak present at low surface coverage at ~20 E-cad shifts to a modestly larger cluster size of ~40 E-cad. This change is presumably due to weak nonspecific interactions between the mutants that support cluster formation at elevated surface coverage. The cluster size distributions of wild-type E-cad exhibit a more dramatic change with increasing surface coverage, particularly in the tails of the distributions. For example, at high and intermediate surface coverage the probability of observing a large cluster (~40 to ~160 E-cad) is significantly increased. This change with increasing surface coverage for wild-type E-cad is likely due to a combination of nonspecific and specific interactions that cause large cluster formation, relative to the cluster formation observed for the mutant.

For kMC simulations, we first turn off specific cis-interactions, so that E-cad can form clusters only through nonspecific lateral interactions. This simulation is used to mimic the system in which the mutant is employed to eliminate specific cis-interactions. The final configuration from a representative simulated trajectory is shown in *Figure 4C*. In addition to E-cad monomers, homogeneously distributed compact clusters formed through nonspecific cis-interactions between mutant E-cad proteins. *Figure 5—figure supplement 1* further shows the cluster size distributions under different on/off rate combinations of the nonspecific interactions. Cluster size distributions can be fitted by a single exponential function $f(N) = Ae^{-N/N_0}$ where $N_0$ corresponds to the characteristic cluster size. *Figure 5—figure supplement 1* indicates that the characteristic cluster size is closely related to the

values of the on and off rates. The simulated on and off rates were therefore optimized so that the cluster size distribution from simulations (red) agreed with the experimental distribution (black) for the cis-mutant (*Figure 5D*). The value of the characteristic cluster size in the experiment was ~29 E-cad, which is equal to the computational characteristic cluster size of ~29 E-cad, within experimental uncertainty. The on and off rates of the nonspecific interaction used to generate the distribution in the simulation are $2 \times 10^5$ s$^{-1}$ and $10^3$ s$^{-1}$, respectively (*Supplementary file 1h*). These on/off *rates* correspond to the effective *rate constants* of $k_{on} \cong 1.1 \times 10^6$ M$^{-1}$s$^{-1}$ and $k_{off} \cong 1 \times 10^3$ s$^{-1}$, based on the calculation developed in our previous studies (*Wang et al., 2018*). These rates correspond to an effective binding affinity in the mM range, for nonspecific cis-interactions.

Subsequently, we carried out simulations in which the specific cis-interaction was turned on. Different combinations of on/off rates for the specific interaction were systematically tested, while the rates of the nonspecific interactions were fixed at the values determined for the cis-mutant. The final configuration from one of these simulations is shown in *Figure 4D*. Relative to the homogeneous and compact clusters observed in the simulations associated with E-cad mutant, the clusters formed when both nonspecific and specific cis-interactions were switched on exhibited extended (linear) configurations. These one-dimensional linear clusters are derived from the polarized cis-binding interface, which is inferred from the x-ray crystal structure of wild-type E-cad (*Harrison et al., 2011*). Cluster size distributions associated with different combinations of on and off rates for specific interactions are shown in *Figure 5—figure supplement 2*. Again, we identified an appropriate combination of specific cis on/off rates that resulted in a similar characteristic cluster size as was observed experimentally for wild-type E-cad, as shown in *Figure 5C*. The value of the characteristic cluster size for the experiment is ~33 E-cad, which is very similar to the computational value of ~34 E-cad from simulations. The on and off rates of the specific interaction that were used to generate the distribution in the simulation are $10^8$ s$^{-1}$ and $10^2$ s$^{-1}$, respectively (*Supplementary file 1h*). These on/off *rates* for the specific cis-interaction correspond to the effective *rate constants* of $k_{on} \cong 2.7 \times 10^6$ M$^{-1}$s$^{-1}$ and $k_{off} \cong 1 \times 10^2$ s$^{-1}$, and to a binding affinity of approximately 10 μM. Comparisons of the specific and nonspecific interactions suggest that the specific cis-binding rate is slightly faster than that of the nonspecific interaction, and the specific cis-interaction is stronger by approximately an order of magnitude.

Finally, in addition to comparisons of cluster size distributions, association time distributions extracted from the simulations were also calculated and qualitatively compared to the experimental association time distributions discussed in the previous section. This ensured that the simulations captured the experimental behavior. *Figure 6* shows the comparison of experimental and simulated association time distributions for mutant and wild-type E-cad. In both simulations and experimental measurements, the association time of E-cad increases when in the presence of specific cis-interactions (wild-type vs. cis-mutant), demonstrating qualitative consistency. We note that the dwell-time distributions from simulations are not necessarily expected to agree quantitatively with experimental measurements, due in part to the difference between the experimental acquisition time (50 ms) and simulation time step (0.01 ns). Notably, the long-time asymptotic behavior of experimental and simulated dwell times have similar behavior (i.e. the slopes of the distribution tails in *Figure 6*), indicating that the simulations accurately capture the salient experimental behavior. Furthermore, experimental phenomena such as desorption, photobleaching, and supported lipid bilayer defects and heterogeneity are not accounted for in the simulations and may limit quantitative comparisons of association times. Overall, these simulation results are qualitatively consistent with longer-lived wild-type E-cad interactions. This is due to specific cis-interactions, as well as to the potential interplay between nonspecific and specific interactions.

## Discussion

An important advance of this research involves the development of a combined experimental and theoretical framework that enables the quantification of lateral binding interactions between proteins confined to fluid, 2D membrane bilayers. The single molecule FRET measurements revealed that both specific and nonspecific cis-interactions contribute to wild-type E-cadherin clustering at a physiologically relevant surface coverage. Complementary kMC simulations provided important insights into the molecular events underlying the FRET distributions, and further extracted rate constants for both specific and nonspecific lateral interactions between the cadherin extracellular

domains. Moreover, these results successfully demonstrated directly that E-cadherin extracellular domains associate through cis-interactions. Prior experimental data supported the role of specific cis-interactions in the assembly of cadherin clusters, both at intercellular adhesions and on supported lipid bilayers at high surface densities (*Harrison et al., 2011*; *Thompson et al., 2019*). However, until recently, direct characterization of E-cad cis-interactions was not possible by traditional methods, due to the weak binding affinity.

Notably, we find that both specific and nonspecific interactions control E-cad clustering on membranes at high surface coverage, and that nonspecific interactions contribute to both mutant E-cad and wild-type E-cad lateral interactions at surface concentrations below the surface coverage threshold for cis-clustering. Although these nonspecific interactions are weaker than specific cis-interactions, they are more frequent, and hence dominate at low concentrations. The conditions employed in these measurements isolated the effects of specific and nonspecific interactions, and they enabled quantitative comparisons with kMC simulations. For both the mutant and wild-type E-cadherin at intermediate surface coverage, where the intermolecular interactions are primarily due to nonspecific interactions, the high-FRET state corresponds to slower diffusion than the low-FRET state. The latter behavior is a result of small, short lived, cluster formation, and was only observable due to the ability to isolate high-FRET objects. However, if one were only able to compare the overall average diffusion of all objects, then the slight decrease in the diffusion coefficient of mutant E-cad at high concentration would not be observable, as previously reported (*Thompson et al., 2019*). We have also shown that for wild-type E-cad, the combination of specific and nonspecific cis-interactions results in the formation of clusters in the range of ~40 to ~160 E-cad, and for the mutant, nonspecific cis-interactions result in an increasing probability of ~40 E-cad clusters. Cell studies have previously reported the formation of clusters of comparable size, independent of trans-interactions. However, we observe larger median cluster size values (*Wu et al., 2015*). This discrepancy could be explained by differences in membrane viscosity, E-cad surface coverage, and/or the dynamic range of cluster size determination techniques.

It was necessary to include both nonspecific and specific interactions in the kMC simulations, in order to accurately reproduce the experimental cluster size distributions. This agreement confirmed the interpretation of the single-molecule FRET data. The rate constants associated with each of these distinct lateral interactions further show that, despite the 10-fold slower dissociation rate of specific cis-bonds, the nonspecific interactions must be taken into account.

The influence of nonspecific interactions on mutant E-cad has not previously been reported. Indeed, it was necessary to combine highly sensitive single-molecule FRET with computational simulations, and to explicitly compare wild-type and cis-mutant E-cad, in order to characterize these weak interactions. Moreover, as these results demonstrate, nonspecific interactions are dynamic and short lived, and would not likely be detected by alternative methods, such as ensemble averaged FRET or photon counting (*Biswas et al., 2015*; *Zhang et al., 2009*). Although nonspecific steric (repulsive) interactions have been invoked to account for membrane protein organization (*Albersdörfer et al., 1997*; *Paszek et al., 2014*; *Qi et al., 2001*; *Schmid et al., 2016*), the potential significance of nonspecific attractive interactions was not fully appreciated prior to this study.

E-cadherin represents a special, and particularly demanding test case for characterizing lateral protein interactions tethered to lipid bilayers, because the cis-bonds have very low affinity and are not detectable in solution. This combination of single molecule FRET and kMC simulations can be extended to other proteins such as nectins that likely interact through higher affinity cis-bonds (*Rikitake et al., 2012*). Although there are approaches for quantifying the 2D trans- (adhesive) affinities and binding rates of membrane receptors, until now, few measurements were able to quantify lateral binding affinities (*Chen et al., 2010*; *Chesla et al., 1998*; *Chien et al., 2008*; *Sarabipour et al., 2015*; *Wu et al., 2008*; *Zhu et al., 2007*), and there are no prior reports of measured off rates. Interestingly, theoretical models of cadherin binding predict cooperativity between trans-binding between opposing cadherins and cis-interactions (*Wu et al., 2010*). The approach described in this study lays the groundwork for directly testing that hypothesis, by comparing cis-binding rates, for example, between cadherins on free membranes versus within adhesion zones.

These findings provided new insights regarding the physical interactions underlying E-cadherin clustering. They also raise the possibility that nonspecific interactions could similarly influence the oligomerization of other membrane proteins. Conversely, the methods described in this study also

open the possibility of quantifying the impact of other factors such as crowding, confinement, or even membrane topography on protein interactions.

# Materials and methods

## Key resources table

| Reagent type (species) or resource | Designation | Source or reference | Identifiers | Additional information |
|---|---|---|---|---|
| Cell line (human) | HEK293T | ATCC, Dr. Keith Johnson, University of Nebraska, Lincoln | CRL-3216 (RRID:CVCL_0063) | authenticated using STR-PCR and tested negative for mycoplasma |
| Transfected construct (human) | CEP 4.2 plasmid | Dr. Lawrence Shapiro, Columbia University | | Encoding hexahistidine-tagged wild-type E-cad and L175D mutant |
| Commercial assay or kit | Alexa Fluor 555 NHS-ester antibody labelling kit | Invitrogen | A20187 | Labeling E-cad |
| Commercial assay or kit | Alexa Fluor 647 NHS-ester antibody labelling kit | Invitrogen | A20186 | Labeling E-cad |
| Chemical compound, drug | DOPC | Sigma-Aldrich | P6354 | |
| Chemical compound, drug | DGS-NTA(Ni) | Avanti Polar Lipids | 790404 | |
| Chemical compound, drug | DOPE-LR | Avanti Polar Lipids | 810150 | |
| Software, algorithm | Custom Matlab-based software | 10.1021/acsmacrolett.8b00004; 10.1021/acsnano.8b02956; 10.1021/acs.jpclett.9b00004 | | Image analysis |
| Software, algorithm | simjFRAP | 10.1038/srep11655 | | Image analysis |

## FRET sample preparation

CEP 4.2 plasmids encoding the hexahistidine-tagged wild-type E-cad and L175D mutant were obtained from Dr. Lawrence Shapiro (Columbia University, NY). The Human Embryonic Kidney 293T (HEK293T) cell line (authenticated using STR-PCR and tested negative for mycoplasma) was from Dr. Keith Johnson (University of Nebraska, Lincoln), where they were purchased from the American Type Culture Collection (Manassas, VA). Cells were cultured in Dulbecco's Minimum Eagle Medium (DMEM) containing 10% fetal bovine serum (FBS) (Life Technologies, Carlsbad, CA) under 5% $CO_2$ atmosphere at 37°C. Cell lines that stably expressed the soluble proteins were generated, by transfecting HEK293T cells with the mutant construct, using Lipofectamine 2000 (Invitrogen, Grand Island, NY) according to the manufacturer's instructions.

HEK293T cell lines that stably expressed hexahistidine-tagged, soluble E-cadherin ectodomains were selected with 200 µg/mL Hygromycin B (Invitrogen). Western blots of the culture medium confirmed protein expression by individual colonies. The colonies that expressed the highest levels of soluble protein were pooled for further protein production. Secreted, hexahistidine-tagged cadherin was then purified from filtered culture medium, by affinity chromatography with an Affigel NTA affinity column, followed by ion-exchange chromatography (Aktapure). Protein purity was assessed by SDS polyacrylamide gel electrophoresis, and the adhesive function was confirmed with bead aggregation assays (*Brieher et al., 1996*).

Purified E-cad extracellular domains with C-terminal 6xHis tags were randomly labeled using an Alexa Fluor 555 (AF555) NHS-ester antibody labeling kit, and both wild-type and L175D mutant were labeled using an Alexa 647 (AF647) NHS-ester antibody labeling kit (succinimidyl ester; Invitrogen, Carlsbad, CA). Protein was reacted with the dye for 1 hr in buffer (25 mM HEPES, 100 mM NaCl, 10 mM KCl, 2 mM $CaCl_2$, 0.05 mM $NiSO_4$, pH 8) at room temperature. Unreacted dye was removed via spin column. Based on absorbance measurements, using extinction coefficients of 150,000 $cm^{-1}$ $M^{-1}$ for the AF555, 239,000 $cm^{-1}$ $M^{-1}$ for the AF647, and 59,860 $cm^{-1}$ $M^{-1}$ for the protein, the labeling stoichiometry was ~1.3 for AF555 labeling of wild-type E-cad and ~2.3 and~1.3 for AF647 labeling of wild-type and mutant E-cad, respectively. A random labeling procedure was selected over a site-specific labeling method so that interactions not necessarily involving the known

cis-interaction interface would still be observed. Functionality of wild-type and mutant E-cad was retained after labeling as indicated by bead aggregation assays (*Brieher et al., 1996*).

1,2-Dioleoyl-sn-glycero-3-phosphocholine (DOPC) was purchased from Millipore Sigma (Burlington, MA). 1,2-dioleoyl-sn-glycero-3-[(N-(5-amino-1-carboxypentyl)iminodiacetic acid)succinyl] (nickel salt) (DGS-NTA(Ni)) and 1,2-dioleoyl-sn-glycero-3-phosphoethanolamine-N-(lissamine rhodamine B sulfonyl) (ammonium salt) (DOPE-LR) were purchased from Avanti Polar Lipids (Alabaster, Alabama). DOPC and DGS-NTA(Ni) were dissolved in chloroform in the molar ratio of 19:1 in a glass culture tube. Following solvent evaporation under a stream of nitrogen, a thin film of lipids was formed on the side of the tube. This lipid film was then hydrated with buffer so the total lipid concentration was 3 mM. This suspension was mixed via vortex and sonicated for 0.5 hr. The vesicles were then extruded through a 50 nm filter membrane (Whatman, Maidstone, UK) 21 times to form unilamellar vesicles with a homogeneous size distribution.

Glass coverslips (Fisher Scientific, Hampton, NH) and fused silica wafers (Mark Optics, Santa Ana, CA) were cleaned with piranha solution for 2 hr and treated by UV-ozone for 0.25 hr. Following surface treatment, the wafers were placed in a custom built flow cell that had been cleaned using Micro-90 detergent solution (International Product Corp., Burlington, NJ). To form supported lipid bilayers, a dispersion of unilamellar vesicles (3 mM total lipid concentration) was carefully injected into the flow cell in order to avoid air bubble formation. Following a 1 hr incubation period, vesicles spontaneously formed a fluid supported lipid bilayer via vesicle fusion (*Cremer and Boxer, 1999*; *Gizeli and Glad, 2004*; *Richter et al., 2006*). Following formation, the bilayer was rinsed with buffer to remove excess vesicles and incubated with 100 mM NiSO$_4$ for 0.5 hr to ensure complete chelation of DGS-NTA(Ni) lipids (*Gizeli and Glad, 2004*; *Nye and Groves, 2008*). The supported lipid bilayer was then exchanged into buffer before injecting 300 μL of a protein buffer solution containing AF555 labeled wild-type E-cad and either AF647 labeled wild-type E-cad and unlabeled wild-type E-cad or AF647 labeled mutant E-cad and unlabeled mutant E-cad, permitting the binding of hexa-histidine-tagged E-cad to the DGS-NTA lipids. In this configuration, the AF555 labeled E-cad served as the FRET donor and the AF647 labeled E-cad served as the FRET acceptor. Two different total wild-type E-cad solution concentrations of $3 \times 10^{-7}$ M and $5 \times 10^{-7}$ M and one total mutant E-cad solution concentration of $5 \times 10^{-7}$ M were studied. *Supplementary file 1c* summarizes the donor and acceptor solution concentrations for the three conditions. The donor concentration was adjusted to allow for single molecule resolution, and the acceptor concentration was optimized to allow for a large number of FRET events, but an insignificant amount of direct excitation of the acceptor. The resulting average donor surface density was ~0.003 E-cad/μm$^2$ for all three experimental conditions. Using the optimized donor and acceptor concentrations, donor bleed-through into the acceptor channel and direct excitation of the acceptor were both determined to be insignificant by imaging control samples containing either donor and unlabeled E-cad or acceptor and unlabeled E-cad and checking for significant emission in the acceptor channel. These control experiments indicated that the FRET signal observed in samples with both donor and acceptor represented physical donor-acceptor interactions. The addition of unlabeled E-cad was necessary in order to reach a surface coverage high enough, such that significant cluster formation had occurred (*Thompson et al., 2019*). This resulted in a large number of high-FRET events, indicated by an acceptor intensity greater than that of the donor. This high surface coverage could not be achieved by only binding donor and acceptor E-cad to the bilayer as this required an extremely high concentration of acceptor, which would result in excessive background emission in the acceptor channel due to direct acceptor excitation by the donor excitation source. All samples were imaged in 25 mM HEPES, 100 mM NaCl, 10 mM KCl, 2 mM CaCl$_2$, 0.05 mM NiSO$_4$, pH eight buffer under

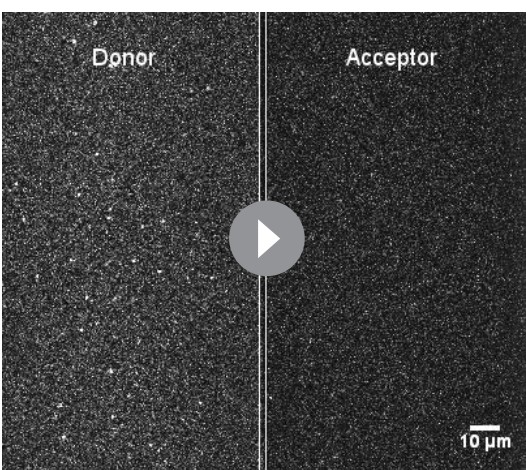

**Video 1.** High-coverage wild-type E-cad movie segment.
https://elifesciences.org/articles/59035#video1

high calcium conditions. An oxygen scavenging system was deemed unnecessary by checking for photo-induced complications in displacement distributions and association time distributions as a function of imaging time (*Figure 1—figure supplements 2–3*).

Control experiments using DOPC/DGS-NTA bilayers without added E-cad and DOPC/DGS-NTA bilayers containing a small fraction of DOPE-LR fluorescent probes were performed to characterize bilayer contamination and lipid diffusion within the supported lipid bilayer, respectively. A low coverage control condition was also tested using wild-type E-cad, where the donor and acceptor concentrations used were the same as the mutant condition, but no unlabeled E-cad was added to confirm that at higher coverage, the FRET signal represented surface coverage dependent interactions. The resulting surface coverage was ~0.2 E-cad/$\mu m^2$, and the apparent average dissociation rate constant was $17.5 \pm 0.6$ s$^{-1}$, nearly an order of magnitude faster than the average dissociation rate constants seen for the high surface coverage conditions. Consistently, the high-FRET dwell times observed at low coverage were drastically shorter than the dwell times observed at higher surface coverages (*Figure 3—figure supplement 1*).

## Single-molecule TIRFM FRET imaging

Imaging of the samples was accomplished using a custom-built prism-based TIRF microscope (Nikon TE-2000 base, 60x water-immersion objective, Nikon, Melville, NY). Custom-built flow cells were mounted on the microscope stage and a 532 nm 50 mW diode-pumped solid state laser (Samba, Cobolt, Solna, Sweden) was used as an excitation source, incident through a hemispherical prism in contact with the wafer on the top of the flow cell. This resulted in an exponentially decaying TIRF field propagating into solution, selectively exciting donor fluorophores at the lipid bilayer-water interface. Fluorescent emissions from the donor and acceptor were separated using an Optosplit III beam splitter (Cairn Research, Faversham, UK) containing a dichroic mirror with a separation wavelength of 610 nm (Chroma, Bellows Falls, VT). Fluorescence from the donor and acceptor were further filtered using a 585/29 bandpass filter and 685/40 bandpass filter (Semrock, Rochester, NY), respectively. The donor and acceptor channels were then projected onto different regions of an Andor iXon3 888 EMCCD camera (Oxford Instruments, Abingdon, UK) maintained at −95°C. An acquisition time of 50 ms was used to capture 12 or 13 image sequences (i.e. movies) of each sample. Three movies were 5 min long and the remaining 9 or 10 movies were 3 min long (see *Videos 1–5* for raw movie segments). Additionally, to allow for accurate donor and acceptor colocalization, the donor and acceptor channels were aligned using images of a glass slide that had been scratched with sand paper, resulting in an irregular alignment image. The details of this image alignment process are described previously (*Faulón Marruecos et al., 2018*). DOPE-LR lipid control experiments were imaged using the same setup for E-cad FRET imaging, except the beam splitter was not necessary and the field of view was allowed to photobleach until the number of DOPE-LR objects was conducive for single-molecule tracking if necessary. Five movies, 5 min in length, were captured for DOPE-LR control experiments using a 50 ms acquisition time.

## Fluorescence recovery after photobleaching (FRAP)

DOPC unilamellar vesicles containing 0.5% DOPE-LR were prepared and used to form a supported lipid bilayer as described previously. SLB incorporated DOPE-LR was bleached by illuminating a circular area of radius ~5 μm with a 532 nm 50 mW diode-pumped solid state laser (Samba, Cobolt, Solna, Sweden) for 4.85 s. After bleaching, DOPE-LR was excited using an Intensilight C-HGFIE lamp (Nikon, Melville, NY). Excitation and emission was separated and filtered using a 532/640 nm TIRF filter cube set (Chroma). The fluorescent emission of DOPE-LR was captured with a Hamamatsu CMOS (ORCA-flash 4.0) camera at an acquisition time of 50 ms. Fluorescent recovery curves were obtained using the ImageJ plug-in simFRAP (*Blumenthal et al., 2015*). *Figure 1—figure supplement 4* shows FRAP recovery snapshots and the FRAP recovery curve, indicating essentially complete recovery and a mobile fraction greater than 0.95.

## Image analysis

All single-molecule movie analysis was performed using custom Matlab-based software, where the methods and algorithms for determining object positions and intensities and linking trajectories have been described elsewhere (*Faulón Marruecos et al., 2018*; *Kienle et al., 2018*). The tracking

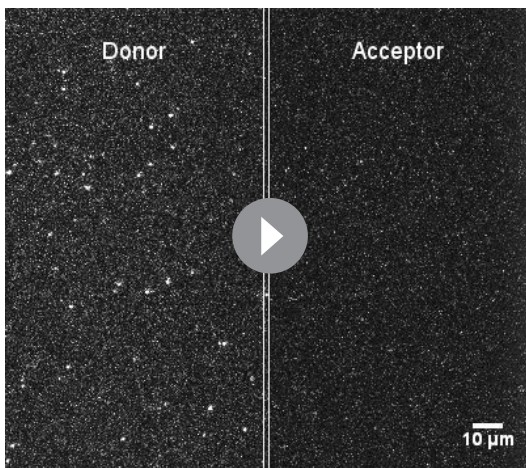

**Video 2.** High-coverage mutant E-cad movie segment.
https://elifesciences.org/articles/59035#video2

software uses established algorithms for localization and tracking, but allows for efficient, integrated analyses of high throughput data, while combining tracking and FRET methods. To briefly summarize, objects that were detected in consecutive frames that were within a user-defined tracking radius (3 pixels or 1.29 μm, for this analysis) were linked into trajectories that could be further analyzed. Object identification was determined using an automated thresholding function that has been described previously (*Kienle and Schwartz, 2019*). This automatic thresholding software allowed for a user-defined number of noise-objects per frame to be identified, as well as the use of a user-defined object radius (0.05 and 1 pixel for this work, respectively). All localized and tracked trajectories longer than two frames from *Video 1* are shown as *Figure 1—figure supplement 5*. Objects that were identified within two pixels in separate channels were identified as a donor-acceptor pair undergoing FRET. A two pixel colocalization distance was selected to allow for potential colocalization between observations with a large position uncertainty, while also allowing for registration error. *Figure 1—figure supplement 6* shows a histogram of position uncertainties for the high surface coverage wild-type condition. As indicated by the distribution, most observations have a position uncertainty well below one pixel (0.43 μm), however the tail of the distribution shows a number of observations with a position uncertainty of approximately one pixel. Therefore, the colocalization distance was set to two pixels and is only applicable when objects were observed within this distance in both channels. Furthermore, the FRET maps (*Figure 1—figure supplement 1*) show two population peaks, both centered around either zero donor intensity or zero acceptor intensity, indicating that colocalization is rare and molecules either exhibit complete energy transfer or zero energy transfer. If the colocalization distance of 2 pixels were too large, resulting in erroneous FRET pair assignment, one would expect to see significant peaks centered around high acceptor and donor intensities. The position of the FRET pair was determined using the object with the greatest signal-to-noise ratio. The FRET state of each object at

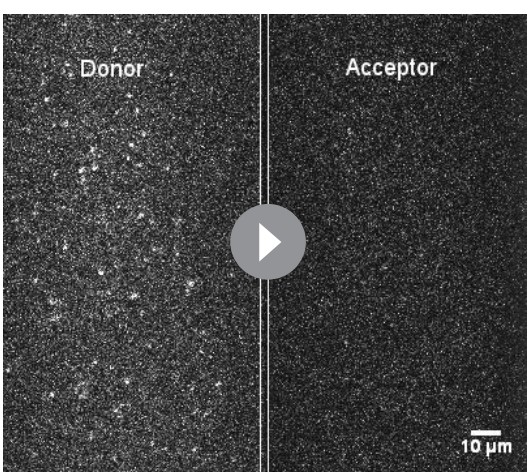

**Video 3.** Intermediate-coverage wild-type E-cad movie segment.
https://elifesciences.org/articles/59035#video3

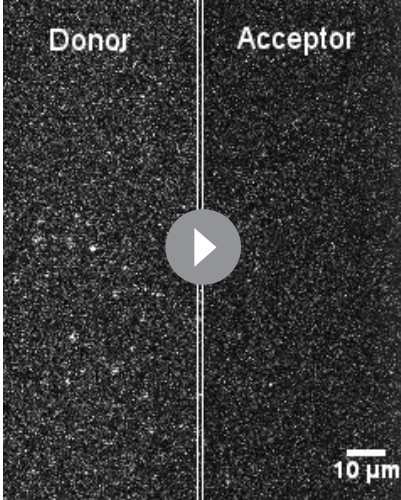

**Video 4.** Low-coverage wild-type E-cad control movie segment.
https://elifesciences.org/articles/59035#video4

every frame was assigned using a method and algorithm described elsewhere (*Chaparro Sosa et al., 2018*). To summarize, two-dimensional heat maps showing the donor intensity ($I_D$) versus acceptor intensity ($I_A$) were constructed. It was apparent that two populations were present at high and low FRET efficiency (*Figure 1—figure supplement 1*). A linear threshold dividing these two populations was calculated by determining the slope and intercept that minimized the integrated heat map values along the dividing line (*Figure 1—figure supplement 1*).

By imaging samples without labeled E-cad, it was apparent that a small number of contaminants were present in the supported lipid bilayer only in the donor channel. These contaminants were generally bright and immobile. Furthermore, due to inherent defects in supported lipid

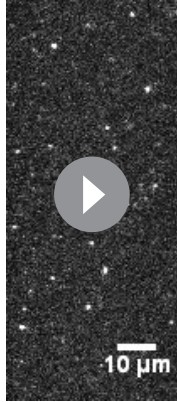

**Video 5.** Lipid tracer control movie segment.
https://elifesciences.org/articles/59035#video5

bilayers, a permanently immobile (or highly confined) population was observed in the donor channel (*Knight et al., 2010*). Traditionally, a displacement-based trajectory filtering procedure or photobleaching is applied to remove these slowly diffusing trajectories in lipid bilayer studies (*Cai et al., 2016*; *Chaparro Sosa et al., 2018*; *Chung et al., 2016*; *Knight and Falke, 2009*; *Knight et al., 2010*; *Ziemba and Falke, 2013*). However, we opted to instead use a median donor intensity trajectory exclusion criterion, as this removed many bright contaminants, donor aggregates, and donor E-cad labeled with multiple fluorophores, but did not accidentally remove slowly diffusing E-cad clusters. A 60th percentile median donor intensity maximum cutoff was selected as this was determined to include single donor E-cad with one fluorophore, while excluding many anomalous trajectories, described above, that were represented by the tail of the median donor intensity distributions (*Figure 1—figure supplement 7*). Intensity-based filtering criteria are frequently used in single-molecule analysis (*Knight and Falke, 2009*; *Knight et al., 2010*). Not using a displacement-based filtering procedure allowed the observation of diffusion over an extremely large dynamic range, which was important here to observe both large clusters and monomers. However, this results in lower than expected average diffusion coefficients, as a small number of apparently immobile trajectories will bypass the intensity exclusion. Because of this, we focus on relative differences in diffusion and do not base any major scientific conclusions on the absolute values of the average diffusion coefficients. To show that our bilayers do in fact exhibit diffusion consistent with previous reports, we have included a short-time diffusion analysis using a displacement-based trajectory filtering procedure (*Figure 2—figure supplement 2*). When this more conventional filtering procedure is applied, we measure average short-time diffusion coefficients within the range seen for supported lipid bilayers for both E-cad and lipids in the bilayer (*Rose et al., 2015*).

For short-time diffusion coefficient determination, only trajectories with a total surface residence time of at least 0.71 s were included, to allow for significant statistical analysis. This surface residence time minimum of 0.71 s was not required for the dissociation rate estimations. Therefore, all trajectories longer than 0.1 s (two frames) were included. Also, trajectories that were observed in the first or last frame were excluded from dwell time and surface residence time analyses to avoid misestimating the time spent in a given state. Lastly, trajectories that lasted longer than 1000 frames were assumed to be contaminants and were removed.

## Surface coverage estimation

The surface coverage in terms of # of E-cad/μm² was estimated according to:

$$\theta = \frac{n_D + n_A - n_{D,AC}}{R_D A_D} \tag{1}$$

where $\theta$ is the surface coverage in terms of # of E-cad/μm², $n_D$ is the number of fluorescent molecules in the donor channel, $n_A$ is the number of fluorescent molecules in the donor channel, $n_{D,AC}$ is

the apparent number of fluorescent molecules in the donor channel for an acceptor control sample that did not contain any donor labeled E-cad, $A_D$ is the area of the donor channel, and $R_D$ is the ratio of donor-labeled protein to total protein. Subtracting the apparent number of fluorescent molecules in the donor channel for a sample without any donor labeled E-cad allowed for the exclusion of contamination in the donor channel, as well as any fluorescence in the donor channel from direct excitation of the acceptor. This estimate assumes a one-to-one transfer of energy from donor to acceptor, complete transfer of energy from donor to acceptor, minimal apparent objects in the acceptor channel that were not actually FRET acceptors, and that labeled and unlabeled E-cad are equally capable of binding to the bilayer. These assumptions were appropriate for these experiments, primarily because the number of objects in the donor channel was much greater than the number of objects in the acceptor channel and because intermediate FRET-states were not significant. Even so, the resulting surface coverage values should be treated as estimates. The fractional surface coverage was averaged over only the first ten frames of each movie to minimize the underestimation of surface coverage due to photobleaching. To further improve estimates, only objects that were tracked for three frames or more were included in surface coverage calculations. This greatly reduced the inclusion of false noise objects that were observed only for one or two frames. These surface coverage values were converted to a fractional areal surface coverage by multiplying by the cross-sectional area of an E-cad extracellular domain, ~9 nm$^2$, assuming the proteins were in an extended conformation due to the presence of calcium (*Lambert et al., 2005*; *Nagar et al., 1996*). Surface coverage estimates are included in *Supplementary file 1d*, both in terms of # of E-cad/µm$^2$ and fractional surface coverage by area, for the three protein solution conditions.

## Average short-time diffusion coefficient determination

All molecular displacements between consecutive frames were separated based on FRET state, and complementary cumulative squared displacement distributions were calculated using histograms of all squared displacements in each of the two states (high and low FRET efficiency), where the squared displacement was defined as the square of the Euclidean distance traveled from frame to frame. Additional distributions were constructed using all molecular displacements from both FRET-states. These distributions were then fitted to a Gaussian mixture model:

$$P\left(R^2 \geq r^2, \Delta t\right) = \sum_{i=1}^{M} c_i e^{-r^2/4\Delta t D_i} \tag{2}$$

where $r$ is the Euclidean displacement between frames, $\Delta t$ is the time between frames (0.05 s), $c_i$ is the fraction of displacements fitted by the $i$th Gaussian term, $D_i$ is the diffusion coefficient for the $i$th term, and $M$ is the number of terms included in the model. These data were satisfactorily modeled by $M = 3$ based upon residual analysis. A three-term Gaussian mixture model was selected because the ability of this model to serve as a robust fitting function to extract an accurate average short-time diffusion coefficient under all conditions, and interpretation of the three diffusive states is strictly avoided. Using the Gaussian mixture model parameters determined from nonlinear fitting, an average short-time diffusion coefficient ($\overline{D}_{short}$) was calculated for both FRET-states and overall:

$$\overline{D}_{short} \sum_{i=1}^{M} c_i D_i \tag{3}$$

where $\overline{D}_{short}$ represented the average diffusion coefficient on the shortest experimentally accessible time-scale.

## Surface residence time distributions

Complementary cumulative residence time (observation time) distributions were constructed for both the high and low-FRET states by separating all trajectories into high-FRET and low-FRET trajectories, where a high-FRET trajectory was defined as any trajectory where the molecule was in the high-FRET state for at least one frame. After trajectory classification, the fraction of molecules that remained on the bilayer a given time after their initial observation ($t_s$) was calculated for both high-FRET and low-FRET trajectories. *Figure 3—figure supplement 2* shows the resulting complementary cumulative surface residence time distributions for the mutant and two wild-type conditions.

## FRET-state dwell time distributions and transition rate determination

Complementary cumulative dwell time distributions were calculated for the two FRET states, corresponding to high and low FRET efficiency, where the apparent dwell time ($\tau$) was defined as the number of consecutive frames a trajectory spent in a given state multiplied by the acquisition time, where the FRET state was determined as described above (*Figure 3—figure supplement 5* and *Figure 3—figure supplement 1*). For these distributions, all dwell times were used, not only dwell times bounded by transitions.

Furthermore, E-cad interactions were modeled using a 3-state Markov model that has been previously used to model protein conformation changes (*Kienle et al., 2018*). To summarize, this model allowed for three states: high-FRET, low-FRET, or off. Therefore, the transition probability matrix had the form:

$$TR = \begin{bmatrix} 1 - p_{LH} - p_{off} & p_{LH} & p_{off} \\ p_{HL} & 1 - p_{HL} - p_{off} & p_{off} \\ 0 & 0 & 1 \end{bmatrix} \tag{4}$$

Where $p_{LH}$, $p_{HL}$, and $p_{off}$ are the probabilities for a transition from the low-FRET state to the high-FRET state, from the high-FRET state to the low-FRET state, and for a trajectory to terminate via photobleaching or desorption, respectively. The value of $p_{off}$ was determined independently by fitting the surface residence times to an exponential distribution. In order to determine the transition probabilities, a maximum likelihood estimate was used based on all trajectory FRET state sequences (assigned as described above). To describe the heterogeneity in these transition probabilities, a likelihood function was defined to allow for beta-distributed transition probabilities. The resulting likelihood function was:

$$LF(S|a_{LH}, b_{LH}, a_{HL}, b_{HL})$$
$$= \prod_k \left[ \frac{B(a_{LH} + N_{LH,k}, b_{LH} + N_{LL,k}) B(a_{HL} + N_{HL,k}, b_{HL} + N_{HH,k})}{B(a_{LH}, b_{LH}) B(a_{HL}, b_{HL})} p_{off}^{N_{off,k}} (1 - p_{off})^{N_{LL,k} + N_{LH,k} + N_{HH,k} + N_{HL,k}} \right] \tag{5}$$

Where $S$ is the sequence of observed FRET states for the $k$th trajectory, B is the beta function, and $N_{HL,k}$, $N_{LH,k}$, $N_{HH,k}$, $N_{off,k}$, and are the number of times within the $k$th trajectory the molecule transitions from the high-FRET state to the low-FRET state, transitions from low-FRET state to the high-FRET state, remains in the low-FRET state, remains in the high-FRET state, and ends, respectively. The model is parameterized by $a_{LH}$, $b_{LH}$, $a_{HL}$, and $b_{HL}$, which are the parameters defining the beta distribution of $p_{LH}$ and $p_{HL}$, respectively. The log of this likelihood function was maximized by iteratively changing the parameters defining the beta distributions describing the transition probabilities between the high and low-FRET states. The average transition rates were then estimated by:

$$r_{LH} = -(\psi(b_{LH}) - \psi(a_{LH} + b_{LH})) / \Delta t \tag{6}$$

$$r_{HL} = -(\psi(b_{HL}) - \psi(a_{HL} + b_{HL})) / \Delta t \tag{7}$$

where $\Delta t$ is the experimental acquisition time, $\psi$ is the digamma function, and $r_{LH}$ and $r_{HL}$ are the average transition rates from the low-FRET state to the high-FRET state and from the high-FRET state to the low-FRET state, respectively. Additionally, for transition from the high-FRET state to the low-FRET state, the average transition rate is equivalent to the average dissociation rate constant ($\bar{k}_d$), since dissociation is a unimolecular reaction. This is not the case for transition from the low-FRET state to the high-FRET state. Resulting beta distributions of state transition probabilities are shown as *Figure 3—figure supplement 6*, and the corresponding probability density functions for state transition rates are shown as *Figure 3—figure supplement 4*. The values of the average transition rates are included in *Supplementary file 1e*. After determining the most likely beta distribution parameters for the transition probabilities, trajectories were simulated using these transition probability distributions and complementary cumulative dwell time distributions were constructed after truncating the simulated trajectories by sampling from the experimental trajectory surface residence time distributions. These theoretical dwell time distributions were compared to the experimental distributions to check for model consistency (*Figure 3—figure supplement 3*).

## Single-molecule TIRFM cluster size distributions

In order to calculate E-cad cluster size distributions, raw trajectory friction factor data were adapted from Thompson, et al. and subjected to further analysis (*Thompson et al., 2019*). Mobility was selected as a means to infer cluster sizes as this allows determination of cluster sizes over a large dynamic range (i.e. greater than two orders of magnitude in diffusion coefficients). Briefly describing the methods used to generate these raw friction factor data: TIRFM was used to observe single AF555 labeled E-cad molecules diffusing on DOPC supported lipid bilayers containing 5% DGS-NTA (Ni) as a function of increasing E-cad surface coverage. Single molecule trajectories were extracted and an effective diffusion coefficient ($D_T$) was calculated for each trajectory according to:

$$D_T = \frac{1}{4T} \sum_{i=1}^{T} \left[ (x_i - x_{i-1})^2 + (y_i - y_{i-1})^2 \right] \tag{8}$$

where $T$ is the duration of the trajectory and $x_i$ and $y_i$ are the Cartesian position coordinates of the trajectory after time $i$. The effective diffusion coefficient was then related to the trajectory friction factor ($f$) by the Einstein relation (*Edward, 1970*):

$$\frac{f}{k_B T} = \frac{1}{D_T} \tag{9}$$

where $k_B$ is the Boltzmann constant, $T$ is temperature, and $D_T$ is the effective diffusion coefficient for a single trajectory. For a more detailed explanation of experimental methods or trajectory friction factor calculations, see *Thompson et al., 2019*.

Considering that mutant E-cad tethered to the bilayer diffuses the same as a single lipid at all surface coverages (*Figure 5—figure supplement 3* and *Thompson et al., 2019*), we can extract the effective size of E-cad clusters assuming additive friction factor contributions from each E-cad molecule in the cluster (*Cai et al., 2016*; *Knight et al., 2010*; *Thompson et al., 2019*; *Ziemba and Falke, 2013*). The apparent trajectory friction factor, $f$, can be expanded as:

$$f = \sum_{i=1}^{N} f_i \tag{10}$$

where $f_i$ is the friction factor contribution due to each E-cad molecule in the cluster and N is the number of protein molecules in the cluster. The friction factor contribution of each protein in the cluster is equal to the friction factor of a lipid in the free-draining limit, assuming bound lipids are well separated and each E-cad molecule tightly binds a single lipid and has minimal contact with additional lipids, all of which are generally true for this system after filtering trajectories (*Knight et al., 2010*). The lipid separation distance for this system should be equal to the diameter of an E-cad extracellular domain, which is approximately 3.4 nm (*Lambert et al., 2005*; *Nagar et al., 1996*). This separation distance is large enough to assume lipid motion is not correlated (*Knight et al., 2010*). Therefore, the trajectory friction factor becomes:

$$f = N f_L \tag{11}$$

Where $f_L$ is the friction factor of an individual lipid. The friction factor of a lipid can be extracted from E-cad trajectory friction factor distributions recognizing that the large peak in the low friction factor limit corresponds to E-cad monomer diffusion. It was determined that $f_L$ = 0.5 s/µm$^2$ corresponded to the friction factor of a lipid (*Figure 5—figure supplement 4*). The apparent cluster size was calculated for each trajectory, and a probability distribution was constructed for each experimental condition.

## Ensemble-time-averaged mean squared displacement

The E-cad trajectory data adapted from Thompson, et al. mentioned above was further compared to trajectory data for fluorescent lipid tracers also from Thompson, et al. to corroborate the claim that E-cad is monovalently bound to a single lipid (*Thompson et al., 2019*). This was done by comparing ensemble-time-averaged mean squared displacement curves between wild-type and mutant E-cad and a lipid in the bilayer. Ensemble-time-averaged mean squared displacement calculation

and fitting for E-cad has been described previously (*Thompson et al., 2019*). Using the raw lipid trajectory data, the ensemble-time-averaged mean squared displacement ($\langle \overline{\delta^2(\tau,T)} \rangle$) was calculated as a function of lag time, $\tau$, according to :

$$\overline{\langle \delta^2(\tau,T) \rangle} = \frac{1}{N} \sum_{i=1}^{N} \frac{\Delta t}{T-\tau} \sum_{n=0}^{\frac{T-\tau}{\Delta t}} \left[ \left( x_{i,n\Delta t + \tau} - x_{i,n\Delta t} \right)^2 + \left( y_{i,n\Delta t + \tau} - y_{i,n\Delta t} \right)^2 \right] \qquad (12)$$

where $N$ is the number of trajectories, $x_{i,\tau}$ and $y_{i,\tau}$ represent the Cartesian position coordinates after time $\tau$, $\Delta t$, is the time between frames (0.06 s), $T$ is the duration of the trajectory, and $\overline{\delta^2(\tau,T)}$ is the time-averaged mean squared displacement of a single trajectory. Only trajectories longer than six frames (0.36 s) were used in mean squared displacement (MSD) calculations and trajectories were truncated at six frames for $\overline{\langle \delta^2(\tau,T) \rangle}$ calculations. Additionally, $\overline{\langle \delta^2(\tau,T) \rangle}$ was only evaluated for lag times where $\frac{t-\tau}{\Delta t}$ was greater than three frames. The $\overline{\langle \delta^2(\tau,T) \rangle}$ lipid curve was then fitted to the Brownian diffusion model (*Meroz and Sokolov, 2015*):

$$\overline{\langle \delta^2(\tau,T) \rangle} = 4 D_{TA} \tau \qquad (13)$$

where $D_{TA}$ represents the time-averaged diffusion coefficient. *Figure 5—figure supplement 3* shows $\overline{\langle \delta^2(\tau,T) \rangle}$ comparisons between wild-type and mutant E-cad at high, intermediate, and low surface coverage values and the resulting values of $D_{TA}$, confirming that E-cad is monovalently bound to a single lipid.

## kMC simulations

### Construction of a domain-based coarse-grained model

Considering the E-cad extracellular regions consisting of five domains (EC1-EC5) (*Harrison et al., 2011*), we constructed a domain-based coarse-grained model to describe the structural arrangement of E-cad proteins. Each E-cad extracellular domain is coarse-grained into a rigid body with a radius of 1.5 nm, and the rigid bodies are spatially aligned into a rod-like shape (*Figure 4A*). These E-cad extracellular domains are further distributed on the plasma membrane, which is represented by the bottom surface of a three-dimensional simulation box. The space above the plasma membrane represents the extracellular region. The extracellular regions of E-cad can form clusters through cis-interactions. Two different types of cis-interactions are considered in the model. The first is the polarized interactions that were observed in the crystal structure. To implement this interaction, we assigned a cis-donor site (purple dots) on the surface of each E-cad N-terminal domain, so that it can bind to a cis-acceptor (red dots) site on the other E-cad. As a result, two adjacent E-cad proteins can be laterally connected through these specific cis-binding interfaces (*Figure 4A*). In addition to the polarized specific interaction, a nonspecific interaction between two E-cads was also considered in the simulation system. As shown in the figure, this interaction can be formed by any pair of two E-cad within a certain distance cutoff. Therefore, it is non-polarized.

### Implementation of the kMC simulation algorithm

Given the surface density of E-cad, an initial configuration is constructed by randomly distributing molecules on the plasma membrane, as shown in *Figure 4B*. Starting from this initial configuration, simulation of the dynamic system is then guided by a kinetic Monte-Carlo algorithm. The algorithm follows a standard diffusion-reaction protocol, as we developed earlier (*Xie et al., 2014a*). Within each simulation time step, stochastic diffusions are first selected for randomly selected E-cad molecules. Translational and rotational movements of the molecules are confined on the surface at the bottom of the simulation box. The amplitude of these movements within each simulation step is determined by the diffusion coefficients of E-cad on a membrane surface. Periodic boundary conditions are implemented such that any E-cad that passes through one side of the cell surface reappears on the opposite side.

In conjunction with diffusion, the reaction associated with nonspecific and specific interactions is triggered stochastically if the binding criteria are satisfied between two E-cad molecules. The specific cis-interactions are triggered by two criteria: (1) the distance between a cis-donor site and a cis-

acceptor site of two molecules is below 1.2 nm cutoff (bond length), and (2) the orientation angles between two monomers are less than 30°, relative to the original configuration of the native E-cad dimer. Nonspecific interactions are triggered by one criterion: the distance between the center of mass of EC1 domains of two E-cad molecules is below 3.2 nm cutoff.

The probability of association is directly calculated by multiplying the on rate of the reaction with the length of the simulation time step. At the same time, dissociations are triggered for any randomly selected interaction with the probability that is calculated by multiplying the off rate of the corresponding reaction with the length of the simulation time step. If an E-cad molecule or E-cad cluster binds to another E-cad, or E-cad cluster through specific or nonspecific binding, they connect and move together subsequently on the surface of the plasma membrane. Finally, the above procedure is iterated until the system evolves into equilibrium patterns in both configurational and compositional spaces.

## Parameter determination in the coarse-grained simulations

The basic simulation parameters, including time step and binding criteria, were adopted from our previous work (*Wang et al., 2018*). The values of these parameters were determined based on benchmark tests in order to optimize the balance between simulation accuracy and computational efficiency. The two-dimensional translational diffusion constant of a single E-cad protein on a lipid bilayer is taken as 10 $\mu m^2$/s and the rotational coefficient as 1° per ns. The values of these parameters were derived from our previous all-atom molecular dynamic simulation results for the diffusions of a cell-surface protein on the lipid bilayer (*Xie et al., 2014b*).

The reaction parameters, including the on and off rates of binding, were chosen from the range that is typical for protein-protein interactions, but at the same time make the simulations computationally accessible. As shown in the next section, the on rates for nonspecific and specific interactions are chosen from the range $10^8$ $s^{-1}$ and $10^4$ $s^{-1}$, corresponding to effective rate constants ranging from $10^4$ $M^{-1}s^{-1}$ to $10^8$ $M^{-1}s^{-1}$. This is a typical range for diffusion-limited rate constants, in which association is guided by complementary electrostatic surfaces at binding interfaces (*Zhou and Bates, 2013*). A wide range of off rates, from $10^4$ $s^{-1}$ and 10 $s^{-1}$, are used to model dissociation of both specific and nonspecific cis-interactions. Therefore, our tests cover the wide range of dissociation constants from milliMolar (mM) to nanoMolar (nM), which is within the typical range for binding of cadherin or other membrane receptors on cell surfaces.

## Sensitivity analysis

To evaluate the sensitivity of different parameters on E-cad clustering, we first performed kMC simulations at different E-cad concentrations (*Supplementary file 1f*). In order to exclude other factors, the on rate and off rates were fixed for nonspecific interactions at $2 \times 10^{-5}$ $s^{-1}$ and $10^3$ $s^{-1}$, respectively, for both mutant and wild-type systems, and the on rate and off rate for specific interactions were fixed at $10^8$ $s^{-1}$ and $10^2$ $s^{-1}$, respectively, for wild-type systems. To build up the initial structure, we assign positions and orientations to 50, 100, 200 E-cad molecules on the membrane surface. The length of each side of the square plasma membrane surface is 400 nm, along both X and Y directions, which gives a total area of 0.16 $\mu m^2$, leading to surface densities of 313 E-cad/$\mu m^2$, 625 E-cad/$\mu m^2$, 1,250 E-cad/$\mu m^2$, respectively. At each concentration, we employed 50 independent replica simulations with random initial seeds. The simulations were extended to 0.8–1.3 s until the average cluster size reached equilibrium, and the final frames of trajectories were used for cluster size analysis. *Figure 5—figure supplement 5* shows resulting cluster size distributions at different concentrations. The solid lines represent one-term exponential fitting for each concentration. For comparison between experimental and simulated characteristic cluster size values, fitting was performed after removal of the data point corresponding to the bin at the smallest cluster size. The positive values of fitted characteristic cluster size (negative slope on semi-logarithmic plot), suggest that small cluster sizes are more favorable than large cluster sizes across the concentration range of 313 E-cad/$\mu m^2$ to 1,250 E-cad/$\mu m^2$. Meanwhile, our results show that large cluster sizes become more populated at higher E-cad concentration for both mutant and wild-type E-cad. This is consistent with experimental results showing that the characteristic cluster size increases with elevating E-cad concentration. Specifically, the characteristic cluster size for the cluster size distribution at 1,250 E-cad/$\mu m^2$ is ~29 E-cad, which is nearly the same for the experimental distribution at a concentration of

1,280 E-cad/$\mu m^2$ (~29 E-cad). These results indicate the robustness of our kMC simulation, suggesting that the clustering configuration generated from the model is sensitive to the total surface coverage of E-cad.

In order to further explore the sensitivity of the model to different binding parameters, we performed smaller kMC simulations involving various on- and off- rates of binding (*Supplementary file 1g*). To fix the surface density at 1,250 E-cad/$\mu m^2$ and accelerate computing speed, we assigned only 50 E-cad molecules on a 100 × 100 $nm^2$ membrane surface. For nonspecific interactions in mutant systems, the on rate values tested were $2 \times 10^6$ $s^{-1}$, $2 \times 10^5$ $s^{-1}$, and $2 \times 10^4$ $s^{-1}$, while the off rate values tested were $10^4$ $s^{-1}$, $10^3$ $s^{-1}$, and $10^2$ $s^{-1}$. For specific interactions in wild-type systems, the on rate values tested were $10^8$ $s^{-1}$, $10^7$ $s^{-1}$, and $10^6$ $s^{-1}$, and the off rate values tested were $10^3$ $s^{-1}$, $10^2$ $s^{-1}$, and $10$ $s^{-1}$, respectively. Simulations were carried out for all different combinations of on/off rates in the mutant system. At each on/off rate, we employed 10 or 20 independent runs with random initial seeds. The simulations were extended to 2 to 4 s, and the final 1 s trajectories were used for cluster size analysis. *Figure 5—figure supplement 1* shows the effects of on/off rate on mutant E-cad cluster size distributions. In each panel, the solid red line represents a single exponential fit, and the values of the characteristic cluster sizes are shown in red. The panels with different on/off ratios have distinct characteristic cluster sizes, while the panels with the same on/off ratio (same binding affinity) have approximately the same characteristic cluster sizes. By comparing simulated and experimental characteristic cluster size values for the mutant, appropriate candidates of nonspecific on/off rates were identified. The optimum nonspecific on and off rates were $2 \times 10^5$ $s^{-1}$ and $10^3$ $s^{-1}$, respectively. Using these nonspecific on/off rates, the wild-type system was simulated using all combinations of specific interaction on/off rates described above. Similarly, *Figure 5—figure supplement 2* shows the effects of specific interaction on/off rates on wild-type E-cad cluster size distributions. In each panel, the solid red line represents the single exponential fit, and the characteristic cluster size value is shown in red. The panels with different on/off rate ratios have distinct characteristic cluster sizes. Finally, analysis of simulated association time distributions can be utilized to select the best candidate from the combinations of specific on/off rates with the same ratio by comparing association time distributions for simulated and experimental trajectories. The selected on/off rates for nonspecific and specific interactions were the only combination of rates that resulted in qualitative agreement between simulated and experimental association time distributions.

## Acknowledgements

This work was supported by the National Institute of General Medical Sciences of the National Institutes of Health under award number 1R01GM117104.

## Additional information

### Funding

| Funder | Grant reference number | Author |
| --- | --- | --- |
| National Institute of General Medical Sciences | 1R01GM117104 | Connor J Thompson<br>Zhaoqian Su<br>Vinh H Vu<br>Yinghao Wu<br>Deborah E Leckband<br>Daniel K Schwartz |

The funders had no role in study design, data collection and interpretation, or the decision to submit the work for publication.

### Author contributions

Connor J Thompson, Zhaoqian Su, Conceptualization, Data curation, Formal analysis, Investigation, Methodology, Writing - original draft, Writing - review and editing; Vinh H Vu, Resources; Yinghao Wu, Deborah E Leckband, Daniel K Schwartz, Conceptualization, Formal analysis, Supervision, Funding acquisition, Investigation, Methodology, Writing - review and editing

### Author ORCIDs

Connor J Thompson  https://orcid.org/0000-0001-6226-7171
Zhaoqian Su  http://orcid.org/0000-0002-8369-0697
Yinghao Wu  http://orcid.org/0000-0003-1181-5670
Daniel K Schwartz  https://orcid.org/0000-0001-5397-7200

### Decision letter and Author response

Decision letter https://doi.org/10.7554/eLife.59035.sa1
Author response https://doi.org/10.7554/eLife.59035.sa2

## Additional files

### Supplementary files

• Supplementary file 1. Additional tables showing experimental parameters, simulation parameters, and sample size values.

• Transparent reporting form

### Data availability

All data generated or analyzed in this work are included in the main text, figure supplements, and Supplementary File 1.

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
