## [Decision Letter]

**Acceptance summary:**

Cadherins are type-I membrane proteins that mediate cell-cell adhesion through both cis and trans interactions. The cis interaction of E-cadherins, crucial for their proper physiological function, is too weak to be measured accurately in solution and was estimated in previous studies to be in the millimolar range. Impressively, Thompson et al. were able to determine both kinetic and dissociation constant characteristics of E-cadherin specific and for the first time the significantly weaker non-specific interactions. They also followed the formation of cis clusters. The approach that was introduced here, single-molecule FRET measurements in combination with kinetic Monte Carlo simulations, could be used to study additional interesting membrane proteins, such as immune receptors.

**Decision letter after peer review:**

Thank you for submitting your article "Cadherin clusters stabilized by a combination of specific and nonspecific Cis-Interactions" for consideration by *eLife*. Your article has been reviewed by three peer reviewers, and the evaluation has been overseen by a Reviewing Editor and Olga Boudker as the Senior Editor. The reviewers have opted to remain anonymous.

The reviewers have discussed the reviews with one another and the Reviewing Editor has drafted this decision to help you prepare a revised submission.

Summary:

Cadherins mediate adhesion between cells and assemble into dense clusters at these adherens junctions. Here, Thompson et al. present a new approach that combines single-molecule FRET measurements with kinetic Monte Carlo simulations in order to measure cis interactions and cis mediated clustering between classical cadherins that are attached to the supported lipid bilayer. The development of such an approach has an important impact and is of interest because, due to experimental difficulties there is a current lack in quantitative experimental measurements of interactions (especially cis interactions) between cadherins and membrane-bound proteins in general. The authors focused on classical cadherins, however, their approach may be applicable to various other membrane-bound proteins. Cadherins are type-I membrane proteins that mediate cell-cell adhesion through both cis and trans interactions. The cis interactions of E-cadherins are too weak to be measured accurately in solution and were estimated in previous studies to be in the millimolar range. Thompson et al. claim to successfully determine both kinetic and dissociation constant characteristics of E-cadherin specific and for the first time the significantly weaker non-specific interactions, and to also follow the formation of cis clusters. They suggest the occurrence of two distinct types of cis-interaction – (i) one that has been described previously based on crystallography data and referred to as specific cis-interaction; and (ii) a second, non-specific interaction that has about 10 times higher dissociation rate compared to the specific cis-interaction, and could also be observed in a cis-mutant E-cadherin.

Opinion:

Regulation of adherens junctions plays many key roles in biology and understanding the physical mechanism of cadherin assembly is certainly important. However, as detailed below, at this point it is unclear to what extent the interesting conclusions are supported by the data. The authors are asked to clarify this and delete or tune down statements that are not fully supported. Many of the issues raised appear to require new experiments. If the authors cannot conduct such experiments they should address these outstanding issues in the manuscript. Because of the significance and number of the issues raised the revised manuscript will be sent for another round of review.

Essential revisions:

Fundamental issues –

1) One of the fundamental issues with the manuscript is the interpretation of the experimental data into suggesting that E-cadherin extracellular domain has a non-specific cis-interaction. What are these non-specific interactions? Are these same for other proteins? Why are they being called non-specific interactions if they occur between two proteins? The authors should consider the fact that the proteins could have formed non-functional aggregates under the current experimental conditions due to changes in their structure such as unfolding. Additionally, the relatively long E-cadherin extracellular domain protein, especially if unfolded, could also interact with the lipid molecules on the membrane or membrane defects including the substrate. Moreover, the poly-His tag can also form cross-links, especially at high densities, due to their interaction with multiple Ni-NTA moieties on the bilayer. Some points to be noted in this regard: (i) The average diffusion coefficient (D) reported here is lower than other reports of protein or lipid D on supported lipid bilayers; (ii) it appears that a large fraction of molecules were found to be immobile and have not been included in the data analysis, (iii) defects in the supported lipid bilayer.

2) Another fundamental issue with the manuscript is the claim of performing single-molecule FRET experiments. The authors have used non-specific labeling of the protein which will likely result in the covalent attachment of fluorophores at multiple positions. In the absence of single-site labeling of individual proteins, it extremely difficult (or impossible) to conclude if an object identified in microscopic images is a single molecule or a larger assembly consisting of more than one molecule.

Technical issues –

Bilayer preparation:

1) Subsection “Nonspecific and Specific Cis-Interactions Are Present in E-cad Clusters”, fifth paragraph: While it is true that D¯short is smaller for high FRET state, the diffusion appears to be quite low for a synthetic supported lipid bilayer.

2) Subsection “FRET Sample Preparation”, last paragraph: 100 mM NiSO_4_ – is this a typo? Otherwise, it is a quite high a concentration of nickel salt. Does this impact the integrity of the supported lipid bilayer?

3) Subsection “Image Analysis”, last paragraph: Did the use of high 100 mM NiSO_4_, as indicated earlier, cause this?

Imaging experiments:

1) Subsection “Nonspecific and Specific Cis-Interactions Are Present in E-cad Clusters”, third paragraph and subsection “Image Analysis”, first paragraph – : A clarification: How were acceptors detected, if they are not associated? Is it that an ROI is used for determining intensities from donor and acceptor channels? If that is the case, then why are the positions different for donor and acceptor molecules. If donor and acceptor molecules were identified independently, then how was it done in the case no FRET? Independent excitation for each channel?

2) Subsection “Nonspecific and Specific Cis-Interactions Are Present in E-cad Clusters”, first paragraph and –subsection “FRET Sample Preparation”, third paragraph: Authors have used non-specific labeling method and may test the functionality of the protein post labeling? This is especially important since the authors claim to observe non-specific interaction between E-cadherin ECD.

3) Subsection “Nonspecific and Specific Cis-Interactions Are Present in E-cad Clusters”: Generally agree with the statement at the start of this paragraph, but curious to know why only two states? Cis association should ideally manifest itself in the form of a continuum of assemblies – monomer, dimer, trimer, higher. Also, FRET is an extremely distance sensitive phenomenon and therefore, if these are large assemblies then why just two FRET states? On the other hand, if these are just dimers, should dimerization result in such drastic changes in the positional displacement of the molecules i.e. diffusion?

4) Subsection “Image Analysis”, first paragraph: Data showing the distribution of the intensities and their ratios would be useful in order to confirm the presence of either just two populations of interacting molecules or the presence of multiple forms of the molecules.

Control experiments:

1) Subsection “FRET Sample Preparation”, first paragraph: The cis-mutant used here is a single point mutation. How about the double mutant, V81DL175D described previously in the following manuscripts?

i) Harrison et al., 2011.

ii) Hong, Troyanovsky and Troyanovsky, 2013.

2) Subsection “FRET Sample Preparation”, last paragraph: The authors may ideally test low density configurations. Reducing the density will not only eliminate any specific cis-interaction but also any non-specific interaction thus, might serve as an internal negative control.

Others:

1) Introduction, first paragraph: Is there any report of lateral interaction between different constituent proteins at the IS? Could authors please list some citations?

2) Introduction, fourth paragraph: PCH is quite sensitive in detecting dimer fraction in a population – it is probably the low cis-interaction affinity and relatively low densities of E-cad-ECD that could have resulted in no cis-interaction between the molecules and therefore, have not been detected in the PCH analysis.

3) Subsection “Nonspecific and Specific Cis-Interactions Are Present in E-cad Clusters”, second paragraph: Is this a citation error? Biswas et al., 2015 manuscript does not appear to have mentioned this value.

Bilayer Preparation –

The method by which the authors form and functionalize their supported lipid bilayers suggests that the bilayers may have a large number of defects and fluorescent artifacts that could complicate analysis. Some points of concern in the preparation methods include:

1) 5% DGS-NTA(Ni) is rather high and will make the supported membranes more prone to defects.

2) Vesicle sonication and extrusion: 30 min sonication is long and the author's make no mention of the sample being on ice or under N2. Lipid oxidation and fragmentation become concerns when lipids are sonicated for this long.

3) Piranha etching: An etching time of 3-5 min is recommended to clean glass surfaces for SLB formation (Lin, W-C, et al., Curr. Protoc. Chem. Biol., 2010 [DOI: 10.1002/9780470559277.ch100131]), Seu, KJ, et al., Biophy. J., 2007 [doi.org/10.1529/biophysj.106.099721]). Longer etch times introduce surface roughness and cause decreased diffusion of lipids in bilayers formed on the substrate (Seu, KJ, et al., Biophy. J., 2007, Figure 2 and Figure 5).

4) Absence of blocking step: Bilayers are commonly blocked with BSA or casein before proteins that will decorate the bilayer are introduced in order to block defects and prevent non-specific sticking of proteins to the glass.

While some bilayer defects are inevitable, or "inherent" as the authors state, the methods the authors use are likely to lead to appreciably more defects than seen in other usages of supported bilayers. It is difficult to assess the degree to which these areas of concern in sample preparation affect image analysis because the authors do not present any video, or even raw images, of the trajectories that they analyze. The reviewers, and presumably future readers as well, would like to see some raw videos corresponding to Figures 1, 2, and 3 with localization and tracking annotations included in their supplementary material so that readers can evaluate the data presented in the main figures. A characterization of the size and number of bilayer defects may also be useful.

His linkage of E-cad to the bilayer –

The authors' use of His6-tagged E-cad and leaving E-cad in solution during imaging may add complications to the experiment and subsequent image analysis that are not thoroughly addressed. While this is certainly dependent on the specific protein, monovalent His6-tag interactions with NTA-Ni lipids unbind with a lifetime of about 8 min (Nye and Groves, 2008). More stably bound proteins likely are in multivalent binding states, and these will become enriched over time. Much of the author's analysis relies on assumptions of monovalent binding. More controls to demonstrate this is the case are necessary.

Controls –

Many experiments lack sufficient controls to corroborate the conclusions that the authors make:

1) The assignment of high-FRET and low-FRET states to clustered and unclustered E-cad, respectively, would be strengthened by accompanying controls of proteins that are known to cluster and known not to cluster. A leucine zipper may be a good control for the high-FRET state.

2) The authors claim that monomers diffuse in the supported lipid bilayer at 0.6 μm^2^/s. This is quite slow for His-conjugated proteins. They also claim that monomers are strictly conjugated to a single lipid. This claim would be substantiated if the diffusion of lipids on these bilayers was also measured, but a strictly monomeric conjugation is unlikely given the conjugation chemistry used. Previously, this group has measured a lipid diffusion of 3 μm^2^/s for a similar lipid composition (Cai et al., 2016). The slow diffusion could very well come from the heavily etched glass. Tracking lipids may help the authors reconcile the unexpectedly slow monomer diffusion.

3) Related to point 2 above, the authors cite Knight et al., 2010to justify usage of a linear scaling relationship between mobility and size in the supported membrane. The Knight et al., paper examines protein domains binding to PI lipids and is not directly relatable to the potentially multivalent His-Tag interactions with Ni-NTA lipids. Especially in light of a significantly slower than usual diffusion coefficients the authors see in their experiments, a substantial amount of extra drag or defects appear to be influencing the cadherin motion. Use of constitutive monomer dimer controls would really help solidify this aspect of the work e.g. as in Chung et al., Biophys. J. 2018 [doi.org/10.1016/j.bpj.2017.10.042].

4) Figure 3: The rates of photobleaching and desorption are not measured, and so it is unclear if they occur on the same time scale as dissociation. The authors state that the curve in the plot represents multiple modes of dissociation, but they could also represent a convolution of photobleaching, desorption, and dissociation on similar time scales. The Materials and methods section states that a 50 mW diode-pumped laser was used as the illumination source, but what was the illumination power at the sample? The rates of photobleaching and desorption should be included as a supplement to this figure and it should be discussed how, if at all, these processes complicate data analysis.

5) WT and mutant E-cad are labeled in a manner that does not control the stoichiometry of the label to protein. The WT and mutant E-cad labeled with 647 have notably different labeling efficiencies (2.3 and 1.3, respectively). How do these differences in labeling efficiencies affect the interpretation of the dwell time distributions, if at all?

6) Related, the donor WT E-cad-AF555 has a labeling efficiency of 1.3. Single E-cad molecules therefore could have 1, 2, or 3 fluorophores. Can the authors distinguish between the dwell of a single donor E-cad molecule with two fluorophores and two diffraction-limited donor E-cad molecules, each with one fluorophore? It seems like with their current labeling scheme, the authors would expect to be sampling multiple populations of dwelling species, even if photobleaching and desorption were corrected for. Because of these complications in how the experiment was conducted, we do not find strong evidence in Figure 3A that cis-interactions exhibit slow dissociation.

7) The authors seem to use diffusion coefficient as the only measurement of E-cad clustering size. Could these measurements be corroborated by a separate measurement of cluster size, say acceptor fluorescence intensity? For example Chung et al., Nature 2010 [doi.org/10.1038/nature08827] used a simple linear scaling assumption for EGFR diffusion as a function of cluster size to claim dimer was the primary species, whereas later single molecule photobleach analysis revealed they were largely higher order oligomers, not well distinguished by mobility (Huang et al., *eLife* 2016 [doi: 10.7554/*eLife*.14107]). Mobility alone, especially in a 2D membrane environment, can be unreliable.

Imaging Experiments –

1) The imaging buffer isn't directly specified. It is left for readers to assume that the imaging buffer is the same as the HEPES buffer that the protein was labeled in. Is this the case? If so, then the lack of an oxygen scavenging system in the buffer may cause complications, such as photo-induced protein crosslinking (Chung et al., 2016). Do the step size distribution and dwell time distribution results change between the 1st and 10th acquisition on the same bilayer? Do results vary with varying laser power?

2) The calcium content of the imaging buffer should also be explicitly noted, as it is important in interpreting the measured diffusion coefficients.

3) The donor is tracked for single molecule tracking and classification of high-FRET and low-FRET states. What is the E-cad-AF555 density for the "low densities" used in single particle tracking experiments?

4) In their Materials and methods section, the authors claim that "due to a combination of bright contaminants and inherent defects in the supported lipid bilayers, a permanently immobile (or highly confined) population was observed in the donor channel." Do the authors know if they have mobile fluorescent contamination? This could be assessed by taking a video with the same acquisition parameters used for step size distribution analysis before adding fluorescent E-cad.

5) Related to 6 (below), could the contaminants be contributing to the FRET signal?

6) FRET analysis is complicated by donor or acceptor molecules that have more than one fluorophore. What steps are taken to filter proteins with more than one dye out of analysis (Hanson, J.A., and Yang, H, J. Phys. Chem. B, 2008 [oi.org/10.1021/jp804440y])?

Image Analysis –

All of the image analysis conducted in this work was built in-house by this research team. While this alone is not an issue, we are concerned the algorithms have not, to our knowledge, been verified with simulations for which the ground truth is known, nor have they been compared to established, verified, and widely-used particle localization and tracking algorithms (Serge A., et al., Nat. Methods 2008 [doi: 10.1038/nmeth.1233]; Jaqaman, K., et al., Nat. Methods 2008 [doi.org/10.1038/nmeth.1237]; Chenouard, N., et al., Nat. Methods 2014 [doi.org/10.1038/nmeth.2808]; Tinevez, J.-Y.,et al., Methods, 2017 [doi.org/10.1016/j.ymeth.2016.09.016]). We have the following questions about the authors' particle localization and tracking algorithms:

1) Do they check for trajectories in which two donors overlap and filter their data to exclude those events from their dwell time distributions or otherwise handle this merging and splitting behavior?

2) Have they quantified their tracking error rate using simulated data for which the ground truth is known? If so, how closely do the simulated data reflect the actual data that the authors have acquired (e.g. in diffusion rate, bilayer density, type of motion)?

3) Have other groups used this algorithm? If so, it would be useful for the authors to cite these papers.

4) The three pixel tracking radius that the algorithm allows seems huge given the pixel size and diffusion coefficients of the species. This large tracking radius could cause complications depending on the density of the donor E-cad. Why was this radius chosen and does it result in tracking artifacts?

5) The 2 pixel, or approximately 800 nm, distance requirement to identify FRET pairs also seems far too big. 800 nm is far out of FRET range of 1-10 nm. We understand that there is some uncertainty in localization, but even so, an 800 nm threshold does not make sense. Papers identifying co-locomotion of two associated membrane proteins often use a much smaller (100 nm) co-localization threshold (Wilkes, S., et al., Science, 2019).

6) Do the authors require that the donor and acceptor signal co-locomote when measuring association times? For how many frames and what distance needs to be maintained to classify the signals as a FRET pair?

7) The step size distribution is built from localizations in adjacent frames. However, depending on the localization algorithm, artifacts can be introduced in a step size distribution if the precision of particle localization is large compared to the size of steps taken. (Cohen, E.A.K., et al. Nat. Comm. 2019 [DOI: 10.1038/s41467-019-08689-x]; Hansen, A.S., *eLife*, 2018 [doi: 10.7554/*eLife*.33125]). We recognize that the authors are most interested in short-time scale diffusion, but the authors can test how accurate the adjacent frame step size distribution is by looking at the distributions from step sizes calculated from every other or every third frame.

Data analysis –

1) The step size distribution is fit with three diffusive states. Why are three states chosen and do the authors have a physical interpretation of what those states are?

2) Can the authors justify why they are interested in the short time diffusion coefficient, D¯short, as opposed to the overall diffusion coefficient, D? Can they provide citations for how this method has been used previously to put their analysis in context?3) In the “Image Analysis” subsection of the Materials and methods section, the authors state that they analyze particle trajectories with median donor intensities in the bottom 60%. Why was the 60% cutoff chosen? Approximately how many fluorophores does this intensity threshold correspond to? What is the probability that they are tracking two closely associated E-cad molecules and not a single molecule?

4) To substantiate their findings the authors may further discuss certain parameter selections as well as their results in the context of current literature and physiological relevance. For example:

a) What is a physiologically relevant surface coverage that is not within cell-cell adhesion? The authors mention ~49,000 E-cad/μm^2^ but within cell-cell adhesion the surface coverage is expected to be significantly higher because of diffusion trap. However, the experiment setting in this paper tests cis interactions independently of trans interaction.

b) We were surprised by the large size of clusters formed without trans interactions. Previous studies have reported much smaller micro clusters – of ~5 molecules if any (Zaidel-Bar lab). Could the authors discuss the cluster sizes observed in their experimental setting.

Hidden Markov Modeling –

1) Work by J. Elf is relevant to the Hidden Markov Modeling performed in this study and should be cited (Persson, J., et al. Nat. Methods, 2013 [DOI: 10.1038/nmeth.2367]).

Figures –

1) Can the authors justify why they chose to plot average dissociation rate constants in Figure 3B? It would be useful to know what rates they extract for the quick "non-specific" interactions and the longer "specific" interactions for each distribution. Based on the text, it seems like they expect this non-specific interaction to be similar for similar bilayer densities of WT and mut E-cad and that the specific interaction to be similar for all bilayer densities of WT E-cad. Is this the case?

---

## [Author Response]

Essential revisions:Fundamental issues –1) One of the fundamental issues with the manuscript is the interpretation of the experimental data into suggesting that E-cadherin extracellular domain has a non-specific cis-interaction. What are these non-specific interactions? Are these same for other proteins? Why are they being called non-specific interactions if they occur between two proteins? The authors should consider the fact that the proteins could have formed non-functional aggregates under the current experimental conditions due to changes in their structure such as unfolding. Additionally, the relatively long E-cadherin extracellular domain protein, especially if unfolded, could also interact with the lipid molecules on the membrane or membrane defects including the substrate. Moreover, the poly-His tag can also form cross-links, especially at high densities, due to their interaction with multiple Ni-NTA moieties on the bilayer. Some points to be noted in this regard: (i) The average diffusion coefficient (D) reported here is lower than other reports of protein or lipid D on supported lipid bilayers; (ii) it appears that a large fraction of molecules were found to be immobile and have not been included in the data analysis, (iii) defects in the supported lipid bilayer.

We appreciate the careful and detailed comments by the reviewers, and we have included additional detail (including new data and characterization), which we believe significantly improves the clarity and robustness of the manuscript.

By “non-specific interactions”, we are referring to interactions that are not related to “specific” molecular-recognition interfaces between E-cad molecules that were observed in crystal structures and tested experimentally (Harrison et al., 2011). These documented, “specific” interactions between cadherin extracellular domains include both trans (opposing) and cis (lateral) bonds. “Non-specific” interactions occur between all proteins, both folded and unfolded, as suggested by the reviewers. In our experiments, the wild-type E-cad also forms “specific” cis-interactions. We have clarified this terminology in the last paragraph of the Introduction section of the revised manuscript. Nonspecific binding between other proteins has frequently been reported (e.g., DOI: https://doi.org/10.1021/bm401302v and https://doi.org/10.1002/prot.25854). Although such attractive interactions are presumably present between all proteins to some extent, they are expected to vary in detail and magnitude, depending on the amino acid composition of the protein surface and on solution conditions. Obviously, since nonspecific interactions are ubiquitous and heterogeneous, it is not possible to describe them in exact terms. However, our findings suggest that they are relatively weak, as generally expected. That said, we cannot completely rule out an additional very weak “specific” bond between the proteins, but there is as of yet no structural evidence for such an interaction or its functional significance.

It is likely that the experimental conditions resulted in the deactivation of some fraction of both wild-type and mutant E-cad. However, we have added function assays using the labeled E-cad, confirming that the labeled E-cad still can perform their adhesive function. A summary of these assays has been added to the subsection “FRET Sample Preparation”. Moreover, by focusing on relative trends and comparisons between wild-type and mutant E-cad, we emphasize the effects of interactions through the specific cis-interface, even in the presence of the effects described by the reviewers. Our dynamic FRET measurements (and the trend observed with surface concentration) are consistent with highly-dynamic and transient surface coverage dependent interactions as opposed to non-functional irreversible aggregation.

In response to a subsequent question below, we describe how we came to the conclusion that E-cad is primarily bound to a single lipid and that multivalent hexahistidine-tag Ni-NTA interactions represent a relatively small population. This was confirmed via MSD analysis (added to this manuscript as Figure 5—figure supplement 3) and short-time diffusion analysis shown previously in Thompson et al., 2019. We also explain why the average short-time diffusion coefficients we report are somewhat lower than other reports of supported lipid bilayer diffusion. As the reviewers know, it is typical for single-molecule observations to be “filtered” by applying criteria to identify and exclude immobile objects and artifacts. Since we are attempting to observe the motion of both individual molecules and large clusters, which move at widely varying rates, it is necessary for us to use less stringent filtering methods than are typically used, so that we do not accidentally exclude slowly moving large clusters. As described below, if we apply more typical filtering criteria, our results are in agreement with those from previous reports.

Based on the reviewers’ comments, we realized that we did not clearly describe the rationale for the use of the 60^th^ percentile median donor intensity cutoff. This was not only used to remove immobile trajectories. Instead, the intensity cutoff served to remove a number of artifactual trajectories, such as irreversible donor aggregates, donors with multiple fluorophores, and bright contaminants. Again, this alternative approach was developed since we were attempting to observe trajectories associated with a large range of cluster sizes, and therefore could not simply use diffusion rate as a filtering criterion, as is commonly done. We have rewritten the paragraph describing the trajectory filtering procedure (subsection “Image analysis”) to more accurately indicate the trajectories excluded and why we choose the trajectory filtering procedure we used.

Lastly, we have added additional characterization experiments to address the question about defects. In particular, we have also characterized the supported lipid bilayers we use via FRAP, demonstrating that the immobile fraction is very small. We have also added references indicating that supported lipid bilayers prepared in the same way as those in this manuscript do not exhibit more defects than is typically observed. Overall, we hope the revisions addressed in detail below regarding E-cad binding to the supported lipid bilayer, diffusion parameters, and supported lipid bilayer quality indicate to the reviewers that these complications do not compromise the integrity of this work.

2) Another fundamental issue with the manuscript is the claim of performing single-molecule FRET experiments. The authors have used non-specific labeling of the protein which will likely result in the covalent attachment of fluorophores at multiple positions. In the absence of single-site labeling of individual proteins, it extremely difficult (or impossible) to conclude if an object identified in microscopic images is a single molecule or a larger assembly consisting of more than one molecule.

We agree that this is an important point and due to the nature of how we labeled donor and acceptor E-cad, it is likely that fluorophores are attached at multiple or different positions for some E-cad molecules. In one regard, this random labeling procedure can be beneficial for detecting nonspecific protein-protein interactions that may not occur in well-defined geometric configurations (i.e. if we labeled E-cad in specific locations to observe specific cis-interactions, we may not observe the nonspecific interactions). Our use of a median donor intensity exclusion criterion removes most donor E-cad with multiple fluorophores and any potential irreversible donor aggregates (Figure 1—figure supplement 7) (this is discussed in detail below). We also use an extremely low concentration of donor E-cad so that the likelihood of observing donor aggregates is very small. This low donor surface coverage is also conducive for single-molecule tracking.

With regards to FRET, we discuss in detail below how we validate the FRET analysis, by robustly identifying distinct FRET states, even though E-cad may have multiple fluorophores. This analysis was briefly described in the original manuscript (and previous related work using this approach was cited). However, we believe that our approach was not clear because it was not explicitly described in detail. This has been remedied in the revised manuscript by the inclusion of an additional figure panel showing a representative “FRET map” (Figure 1A). Below we also describe how it was determined that E-cad is primarily monovalently bound to a single lipid and these data have been added to the revised manuscript. The fact that E-cad diffuses at the same rate as a single lipid is consistent with the observation of single molecules. In the revised manuscript, we have also included multiple segments from raw videos (Videos 1-5), which may provide additional visual evidence of single molecule observations for expert practitioners.

Technical issues –Bilayer preparation:1) –Subsection “Nonspecific and Specific Cis-Interactions Are Present in E-cad Clusters”, fifth paragraph: While it is true that D¯short is smaller for high FRET state, the diffusion appears to be quite low for a synthetic supported lipid bilayer.

We thank the reviewer for pointing out the need for discussion of this apparent discrepancy. As described briefly above, the reason why the average diffusion coefficient reported in our manuscript is lower than that reported in other reports is primarily due to the trajectory filtering procedure we apply, which is required in our experiments because we are attempting to observe a large dynamic range of diffusion coefficients. Numerous single molecule studies involving supported lipid bilayers directly remove slow moving or immobile molecules based on a displacement exclusion criterion or by photobleaching the field of view and allowing highly-mobile molecules to diffuse back in the field of view before imaging (see papers: https://doi.org/10.1016/j.bpj.2010.08.046, https://doi.org/10.1016/j.bpj.2008.10.020, https://doi.org/10.1016/j.bpj.2016.10.037, https://doi.org/10.1021/acsami.8b05523, https://doi.org/10.1021/jacs.5b12648, https://doi.org/10.1016/j.chemphyslip.2013.04.005). We do not use such a trajectory exclusion criterion here as it has the potential to exclude trajectories of interest corresponding to large, slowly diffusing, lateral clusters. Instead, we require the ability to observe trajectories diffusing over an extremely large dynamic range of greater than two orders of magnitude. So instead of using an exclusion criterion based on displacement, we apply an indirect criterion to remove anomalous trajectories that uses intensity. Inherently, this filtering procedure does not remove all immobile trajectories that pass the intensity exclusion criterion, but it successfully removes irreversible donor aggregates, donor E-cad with multiple fluorophores, and a number of contaminants present in the bilayer that were determined to be inherently brighter than a single fluorophore by imaging control samples without labeled E-cad. A similar intensity exclusion criterion is frequently used to remove donor aggregates and anomalous trajectories in single-molecule tracking experiments (DOIs: https://doi.org/10.1016/j.bpj.2010.08.046, https://doi.org/10.1016/j.bpj.2008.10.020).

To demonstrate that our supported lipid bilayers do exhibit diffusion consistent with previous reports when subjected to similar trajectory filtering, we have tracked singly labeled lipids in the supported lipid bilayer using the same imaging system used for FRET samples and reanalyzed all E-cad FRET data using a displacement removal criterion similar to that typically used. (we removed trajectories with a median step displacement less than 0.3 µm). As expected, this resulted in significantly faster average diffusion coefficients for all conditions. The resulting overall complementary cumulative squared displacement distributions have been added as Figure 2—figure supplement 2A and the resulting values of D¯short are included as Figure 2—figure supplement 2B. It is apparent that diffusion is similar for the mutant E-cad, the intermediate surface coverage wild-type E-cad, and for a lipid in the bilayer, but slightly slower for the high surface coverage wild-type condition, presumably because of the presence of clusters. All values of D¯short (~1.2 µm^2^/s) are within a reasonable range seen for synthetic supported lipid bilayers (https://doi.org/10.3390/membranes5040702). Furthermore, inspection of the trajectory averaged friction factor distributions (Figure 5—figure supplement 4), shows a peak at ~0.5 s/µm^2^ (i.e. a diffusion coefficient of ~2 µm^2^/s) at all conditions. Based on this analysis, we conclude that the primary E-cad population we observe exhibits diffusion as expected, but that our trajectory filtering method allows observation of both extremely mobile and very slowly diffusing trajectories. This is a subtle but important point, and we thank the reviewers for encouraging us to describe these methods in greater detail, and to include additional control data.

We have added additional discussion to the second paragraph of the “Image Analysis” subsection in the Materials and methods to clarify why we choose the trajectory filtering method based upon intensity, and we refer the reader to the added diffusion analysis which uses a standard removal of slow or confined trajectories. In this paragraph we also have added references to previous works that use displacement-based trajectory filtering. Additionally, we have tracked labeled lipid probes in E-cad-free bilayers, using the same experimental setup as for FRET samples. A description of these additional control experiments and the results were added to the subsection “FRET Sample Preparation”.

While our trajectory filtering method results in an average diffusion coefficient that is lower than often reported, this does not influence the main findings of the manuscript. Importantly, we focus on the relative differences and trends between samples that must be due to E-cad clustering. Previously, we used a similar trajectory filtering procedure based upon position uncertainty, in order to test for cadherin lateral clustering on supported lipid bilayers over a wide range of surface coverage values. The latter analyses used diffusion coefficients, anomalous diffusion exponents, surface residence times, and friction factor distributions (DOI: https://doi.org/10.1021/acs.jpclett.9b01500). In our previous work, we compared wild-type E-cad to cis-mutant E-cad behavior, as a function of surface coverage. The relative comparison to the cis-mutant E-cad enabled the quantitative determination of cis-interaction-mediated lateral clustering that simply would not have been apparent if we had removed all slowly diffusing trajectories.

2) Subsection “FRET Sample Preparation”, last paragraph: 100 mM NiSO_4_ – is this a typo? Otherwise, it is a quite high a concentration of nickel salt. Does this impact the integrity of the supported lipid bilayer?

We thank the reviewer for pointing out the need to provide evidence that this nickel solution does not impact the bilayer integrity. This concentration was selected initially based on the results from Gizeli and Glad, 2004, where a 5% DGS-NTA fluid supported lipid bilayer was successfully formed via vesicle fusion and was rinsed with a concentrated nickel solution, in order to saturate all NTA lipids. Similarly, a 100 mM nickel solution has elsewhere been used by Nye and Groves, 2008. Therefore, it is not expected that rinsing the bilayer with this nickel solution alters the bilayer integrity. A reference to these two manuscripts has been added to the manuscript where the nickel rinse step is described in the subsection “FRET Sample Preparation”.

3) Subsection “Image Analysis”, last paragraph: Did the use of high 100 mM NiSO_4_, as indicated earlier, cause this?

As noted above, it is not expected that the nickel solution rinse compromised the bilayer integrity as this is a frequent step for histag-NTA binding studies. A small number of bright, primarily immobile, contaminants were observed in the supported lipid bilayers, prior to the addition of labeled E-cad. Also, potential donor aggregates and a small number of inherent bilayer defects were detected and associated trajectories were primarily removed from further analysis. We have rewritten this paragraph (subsection “Image Analysis”) describing trajectory filtering, in order to more accurately reflect the anomalous trajectories observed and clarify how they were handled.

Imaging experiments:1) –Subsection “Nonspecific and Specific Cis-Interactions Are Present in E-cad Clusters”, third paragraph and subsection “Image Analysis”, first paragraph: A clarification: How were acceptors detected, if they are not associated? Is it that an ROI is used for determining intensities from donor and acceptor channels? If that is the case, then why are the positions different for donor and acceptor molecules. If donor and acceptor molecules were identified independently, then how was it done in the case no FRET? Independent excitation for each channel?

Acceptors were only detected if they were associated with a donor E-cad and if they were excited through FRET with the donor fluorophore (i.e. a ROI was used for determining intensities from donor and acceptor channels, as you mention). We have included multiple raw videos showing the donor and acceptor channels as Videos 1-5, which may help clarify this issue. Independent excitation of the donor and acceptor was not performed in this case, since acceptor-labeled E-cad was not generally added at low enough concentrations to allow localization. For this reason, among others described in the Results section (e.g. the subsection “Nonspecific and Specific Cis-Interactions Are Present in E-cad Clusters”), we cannot directly determine when a donor E-cad is a monomer. Because of this, we take great care to avoid any discussion of interaction association kinetics for the single-molecule FRET results. Instead, we focus on dissociation kinetics, as we know an acceptor E-cad must be associated with a donor E-cad if we observe a high-FRET efficiency (Figure 1A and Figure 1—figure supplement 1).

The actual physical positions are not different for a donor and acceptor FRET pair. We typically do not see objects with intensities above background levels in both the donor and acceptor channels within the colocalization radius in this system, but in the rare cases where this is observed, the position of the FRET pair is tracked using the position of the donor or acceptor with the greatest signal to noise ratio. The 2-pixel colocalization radius was necessary because of both the position uncertainty and registration error. Given the low concentration of donor E-cad, this is not expected to be a concern. We have added additional clarification on this parameter in the “Image Analysis” subsection.

2) Subsection “Nonspecific and Specific Cis-Interactions Are Present in E-cad Clusters”, first paragraph and –subsection “FRET Sample Preparation”, third paragraph: Authors have used non-specific labeling method and may test the functionality of the protein post labeling? This is especially important since the authors claim to observe non-specific interaction between E-cadherin ECD.

We thank the reviewer for this suggestion. We have tested the functionality of labeled E-cad via bead aggregation assays. These assays indicated that the labeled E-cad retained their adhesive function and did not show a clear difference between unlabeled E-cad. These results have been added to the Materials and methods subsection “FRET Sample Preparation”.

3) Subsection “Nonspecific and Specific Cis-Interactions Are Present in E-cad Clusters”: Generally agree with the statement at the start of this paragraph, but curious to know why only two states? Cis association should ideally manifest itself in the form of a continuum of assemblies – monomer, dimer, trimer, higher. Also, FRET is an extremely distance sensitive phenomenon and therefore, if these are large assemblies then why just two FRET states? On the other hand, if these are just dimers, should dimerization result in such drastic changes in the positional displacement of the molecules i.e. diffusion?

Thank you for pointing out the need to clarify why we considered only two FRET states. We expect that there is a continuum of assemblies due to cis-association, as suggested by the reviewer. However, we cannot distinguish between these assemblies via FRET for multiple reasons, and we can only distinguish the association of a donor E-cad with one or more acceptor E-cad molecules, i.e. only one FRET state is distinguishable. To help visualize this, we have added a representative “FRET map” as Figure 1A (and additional examples in Figure 1—figure supplement 1). The representative FRET map shown in Figure 1A shows two apparent FRET states indicated by the asterisks at high donor intensity and zero acceptor intensity (low-FRET state) and the peak at high acceptor intensity and zero donor intensity (high-FRET or associated state). These two states correspond to either negligible energy transfer (low-FRET) or nearly complete energy transfer (high-FRET). As described in the text, a simple criterion is applied to divide the map into high and low FRET regions, which is used in subsequent analysis. As seen in the figure, it would not be straightforward to subdivide the high-FRET peak into distinct populations. Given this limitation, we use FRET as an indicator of when a donor E-cad is associated with an acceptor E-cad (as part of a cluster of indeterminate size). We use the time a trajectory spends in the high-FRET state to determine the interaction dissociation kinetics, since the time in the high-FRET state is presumed to correspond to cis-association. The high-FRET state will include a continuum of assemblies, but we cannot differentiate between these microstates. In order to determine the average dissociation rate, it is not necessary to do so. We have added a sentence in the first paragraph of the Results section that describes the two populations observed and refers the reader to a representative FRET map (Figure 1A) and to the FRET maps added as Figure 1—figure supplement 1 to better illustrate what is meant by two FRET states.

While this analysis was described briefly in the original text (and previous papers that used this approach to identify populations in intermolecular and intramolecular SM-FRET experiments were cited) we believe that the inclusion of a representative FRET map will be very helpful for the reader to understand the approach, and we thank the reviewers for prompting us to do so.

4) –Subsection “Image Analysis”, first paragraph: Data showing the distribution of the intensities and their ratios would be useful in order to confirm the presence of either just two populations of interacting molecules or the presence of multiple forms of the molecules.

As suggested, we have added a representative two-dimensional FRET heat map to the main text (Figure 1A) and additional FRET maps as Figure 1—figure supplement 1. The FRET maps indicate two discrete populations at high- and low-FRET efficiency. The population at low-FRET efficiency represents donor E-cad that are not associated with acceptor E-cad, and the population at high-FRET efficiency corresponds to acceptor E-cad that are associated with donor E-cad. The location of the two population peaks indicates either complete energy transfer or negligible energy transfer, with no intermediate states.

Control experiments:1) Subsection “FRET Sample Preparation”, first paragraph: The cis-mutant used here is a single point mutation. How about the double mutant, V81DL175D described previously in the following manuscripts?i) Harrison et al., 2011.ii) Hong, Troyanovsky and Troyanovsky, 2013.

We thank the reviewer for suggesting the double cis-mutant. The single point mutant was selected for this study because we previously determined that this single point mutation is sufficient to largely abolish cis-clustering on a supported lipid bilayer (DOI: https://doi.org/10.1021/acs.jpclett.9b01500). Cis-interactions involve hydrophobic interactions between residues V81 and L175 and Harrison et al., 2011, showed that either the V81D or L175D mutation successfully disrupts cis-interactions in the crystal structure as intended. Therefore, mutating either side of the asymmetric cis-interaction interface is believed to be sufficient based on multiple literature reports.

2) –Subsection “FRET Sample Preparation”, last paragraph: The authors may ideally test low density configurations. Reducing the density will not only eliminate any specific cis-interaction but also any non-specific interaction thus, might serve as an internal negative control.

This is a good suggestion for a negative control, and we have performed additional experiments as suggested. In particular, we have tested an additional low surface coverage wild-type E-cad condition using the same donor and acceptor concentrations as the mutant condition, but without any unlabeled E-cad that was used to elevate the surface coverage for the original three conditions. The surface coverage for this low coverage control was ~0.2 E-cad/µm^2^. We calculated the average dissociation rate constant for any apparent interactions that may be present at this low surface coverage, by using the three-state Markov model. The resulting value of the average dissociation rate constant was 17.5 ± 0.6 s^-1^, nearly an order of magnitude faster than the average dissociation rate constants seen for the high surface coverage conditions. Consistently, the high-FRET dwell times were extremely short (Figure 3—figure supplement 1). This implies that the association events observed at this low surface coverage were extremely short-lived and difficult to distinguish from noise. Overall, this additional control shows that the FRET signal we observe at high surface coverage does represent surface coverage-dependent binding. This result is also consistent with results from previous work, which showed negligible cluster formation at low surface concentration. We have added a discussion of this additional control condition in the Materials and methods subsection titled “FRET Sample Preparation”, and we have added the high-FRET dwell time distribution as Figure 3—figure supplement 1 for comparison with other dwell time distributions.

Others:1) Introduction, first paragraph: Is there any report of lateral interaction between different constituent proteins at the IS? Could authors please list some citations?

We have added a citation to a review that discusses lateral interactions at the IS to the revised manuscript in the Introduction.

2) Introduction, fourth paragraph: PCH is quite sensitive in detecting dimer fraction in a population – it is probably the low cis-interaction affinity and relatively low densities of E-cad-ECD that could have resulted in no cis-interaction between the molecules and therefore, have not been detected in the PCH analysis.

We thank the reviewer for pointing this out. This sentence in the Introduction has been corrected to indicate that the expected reason why PCH was unable to detect cis-interactions was the relatively low densities of E-cad studied.

3) Subsection “Nonspecific and Specific Cis-Interactions Are Present in E-cad Clusters”, second paragraph: Is this a citation error? Biswas et al., 2015 manuscript does not appear to have mentioned this value.

This was a citation error and it has been corrected in the subsection “Nonspecific and Specific Cis-Interactions Are Present in E-cad Clusters”. Thank you for pointing it out.

Bilayer Preparation –The method by which the authors form and functionalize their supported lipid bilayers suggests that the bilayers may have a large number of defects and fluorescent artifacts that could complicate analysis. Some points of concern in the preparation methods include:1) 5% DGS-NTA(Ni) is rather high and will make the supported membranes more prone to defects.

5% DGS-NTA(Ni) was selected based on the results from Gizeli and Glad, 2004, as mentioned above. They directly observed DOPC/DGS-NTA(Ni) supported lipid bilayer formation via vesicle fusion using a frequency acoustic waveguide device and optimized the fraction of DGS-NTA(Ni). They determined that 5% DGS-NTA(Ni) successfully formed a supported lipid bilayer via vesicle fusion, but that 10% did not. The fluidity of the 5% DGS-NTA(Ni) supported bilayer was then confirmed via FRAP. Based on their results, we selected 5% DGS-NTA(Ni) to allow for a high surface coverage of E-cad, while still forming a fluid supported lipid bilayer via vesicle fusion. We have added an additional reference to this paper when describing forming a supported lipid bilayer via vesicle fusion in the subsection “FRET Sample Preparation”. Additionally, it also has elsewhere been reported that supported lipid bilayers can be formed with up to 10% DGS-NTA(NI) without noticeable loss of bilayer quality (DOI: https://doi.org/10.1021/la703788h). Other studies have successfully used supported lipid bilayers with similarly high DGS-NTA(Ni) fractions, such as https://doi.org/10.1073/pnas.1513775112. Thus, we do not believe the 5% DGS-NTA(Ni) lipid used in supported lipid bilayers compromised the bilayer integrity.

2) Vesicle sonication and extrusion: 30 min sonication is long and the author's make no mention of the sample being on ice or under N2. Lipid oxidation and fragmentation become concerns when lipids are sonicated for this long.

We have previously determined that, in our hands, sonicating vesicles for 30 min apparently does not compromise supported lipid bilayer integrity (DOIs: https://doi.org/10.1021/acsami.8b05523 and https://doi.org/10.1016/j.bpj.2016.10.037) based on standard characterization methods. Perhaps, if a lipid more susceptible to oxidation was used, oxidation would be significant.

Nevertheless, to confirm the formation of a continuous supported lipid bilayer via vesicle fusion using a 30-minute sonication time and 2-hour piranha cleaning time, we have added FRAP results to this manuscript for a DOPC supported lipid bilayer formed using these preparation times. The results have been added to the manuscript as Figure 1—figure supplement 4. These results indicate that a 30-minute vesicle sonication time and 2-hour coverslip piranha etch do not compromise the resulting supported lipid bilayer, as the bilayer exhibits nearly full recovery after bleaching with a mobile fraction greater than 0.95. A description of FRAP methods and a summary of the results has also been added to the Materials and methods section.

3) Piranha etching: An etching time of 3-5 min is recommended to clean glass surfaces for SLB formation (Lin, W-C, et al., Curr. Protoc. Chem. Biol., 2010 [DOI: 10.1002/9780470559277.ch100131]), Seu, KJ, et al., Biophy. J., 2007 [doi.org/10.1529/biophysj.106.099721]). Longer etch times introduce surface roughness and cause decreased diffusion of lipids in bilayers formed on the substrate (Seu, KJ, et al., Biophy. J., 2007, Figure 2 and Figure 5).

While extended piranha etching times may cause modestly decreased diffusion of lipids in a supported lipid bilayer on the etched substrate, extended piranha cleaning times are used extensively for supported lipid bilayer studies and apparently do not result in inherently poor-quality supported lipid bilayers (DOIs: https://doi.org/10.1016/j.chemphyslip.2013.04.005, https://doi.org/10.1016/j.bpj.2008.10.020, https://doi.org/10.1021/acsami.8b05523, https://doi.org/10.1016/j.bpj.2016.10.037, https://doi.org/10.1016/j.bpj.2010.08.046, https://doi.org/10.1016/j.bpj.2018.04.019). Furthermore, the manuscript referenced by the reviewer (doi.org/10.1529/biophysj.106.099721) suggests using the coverslip cleaning method and duration as a means to control bilayer fluidity, without changing the bilayer composition.

As mentioned in the response to the comment directly above, to verify that extended etching times did not compromise supported bilayer integrity, we used FRAP to determine the supported lipid bilayer mobile fraction of a DOPC supported lipid bilayer formed using vesicles that had been sonicated for 30 minutes on a coverslip that had been piranha cleaned for 2 hours. The results are shown as Figure 1—figure supplement 4 and indicate the bilayer had a mobile fraction greater than 0.95 and exhibited nearly complete recovery. Therefore, we believe that the extended piranha etching time employed does not cause poor supported lipid bilayer formation and does not compromise the integrity of this work.

4) Absence of blocking step: Bilayers are commonly blocked with BSA or casein before proteins that will decorate the bilayer are introduced in order to block defects and prevent non-specific sticking of proteins to the glass.

As suggested by the reviewer, the use of a blocking agent can be helpful in certain experiments, such as for measurements of specific adsorption to surface (i.e. bilayer) binding sites. However, for these studies we chose not to use a blocking agent. This is because we did not want BSA or another blocking protein present on the supported bilayer, because small amounts of residual blocking agent could affect both the E-cad interactions and diffusion.

Therefore, instead of using blocking agents to reduce the prevalence and influence of residual defects, we focused on preparing high quality bilayers and the removal of artifacts via the use of described criteria used for trajectory analysis. Moreover, we focused on relative comparisons of diffusion and trends between samples over a wide dynamic range of diffusion coefficients. Also, as indicated by FRAP (Figure 1—figure supplement 4) and previous reports of bilayers formed according to similar procedures, the supported lipid bilayers used here are largely continuous and free of overwhelming defects. Therefore, we believe that the use of a blocking agent is not required and the potential complications due to residual blocking proteins would complicate the data interpretations and main conclusions of the manuscript.

In reviewing the manuscript, we understand that the final paragraph of the “Image Analysis” section in the original submission was misleading and could be interpreted to mean that the supported lipid bilayers used in our experiments exhibited a large number of defects, and that the 60^th^ percentile median intensity cutoff was used primarily to remove anomalous trajectories due to these defects. This is not the case, and as described above, this paragraph (subsection “Image Analysis”) has been rewritten to more accurately state why the intensity cutoff was used.

While some bilayer defects are inevitable, or "inherent" as the authors state, the methods the authors use are likely to lead to appreciably more defects than seen in other usages of supported bilayers. It is difficult to assess the degree to which these areas of concern in sample preparation affect image analysis because the authors do not present any videos, or even raw images, of the trajectories that they analyze. The reviewers, and presumably future readers as well, would like to see some raw videos corresponding to Figures 1, 2, and 3 with localization and tracking annotations included in their supplementary material so that readers can evaluate the data presented in the main figures. A characterization of the size and number of bilayer defects may also be useful.

We thank the reviewer for this suggestion. We have included five representative raw video segments with the revised manuscript as Videos 1-5. One video is included for each of the three high surface coverage conditions and a fourth video is included from the low coverage control experiment. The fifth video is included from the single-channel lipid control experiment. Furthermore, we have included the localized and tracked trajectories from the high surface coverage wild-type condition video segment as Figure 1—figure supplement 5. Lastly, as mentioned above, we have characterized the continuity of the supported lipid bilayers we used by adding FRAP results indicating a supported lipid bilayer mobile fraction greater than 0.95, suggesting that the bilayer defects affect less than 5% of the bilayer surface (Figure 1—figure supplement 4 and subsections “Fluorescence Recovery After Photobleaching (FRAP)” and “Image Analysis”).

His linkage of E-cad to the bilayer –The authors' use of His6-tagged E-cad and leaving E-cad in solution during imaging may add complications to the experiment and subsequent image analysis that are not thoroughly addressed. While this is certainly dependent on the specific protein, monovalent His6-tag interactions with NTA-Ni lipids unbind with a lifetime of about 8 min (Nye and Groves, 2008). More stably bound proteins likely are in multivalent binding states, and these will become enriched over time. Much of the author's analysis relies on assumptions of monovalent binding. More controls to demonstrate this is the case are necessary.

We thank the reviewer for pointing out the need to discuss potential multivalent binding of E-cad to the supported lipid bilayer. As described below, we have included additional control experiments to support the assumption of primarily monovalent binding. However, we would like to note that this assumption is related only to the analysis of cluster sizes, and does not influence the FRET analysis of binding times, which are the main focus of the manuscript.

To directly corroborate that E-cad is primarily monovalently bound to a single lipid in our particular experiments, we have added an ensemble-time-averaged MSD analysis comparing the diffusion of labeled lipids in the bilayer and wild-type and mutant E-cad as a function of surface coverage to the revised manuscript (Figure 5—figure supplement 3 and subsection “Single-Molecule TIRFM Cluster Size Distributions”) using the data from Thompson et al., 2019. We addressed this question extensively in that previous work, showing for example, that the diffusion of mutant E-cad was identical to that of a lipid across a large range of surface coverage (and that wild-type E-cad had the same diffusion coefficient at low surface coverage, where clustering was negligible). Most importantly, the lipid MSD is indistinguishable from the mutant MSDs at all three surface coverages employed in this manuscript and the resulting ensemble-time-averaged diffusion coefficients are equivalent within experimental error. This confirms that E-cad is primarily monovalently bound to a single lipid, and we thank for reviewer for suggesting this additional control experiment. We recognize that this may be an issue in some experiments, including those referenced in the paper cited by the reviewer, which employed decahistidine tags. Even in our experiments, it is probable that a small population of E-cad is multivalently bound. For this reason, we primarily focus on the change of the cluster size distributions and the relative difference between wild-type and mutant E-cad, and our main conclusions do not rely on the assumption of purely monovalent binding. In the subsection “Heterogeneous kMC Simulations Differentiate Specific and Nonspecific Interactions”, we have added that we are primarily focused on relative changes in cluster size distributions.

We have rewritten the Materials and methods section describing the cluster size calculations to more accurately describe how we concluded E-cad is primarily conjugated to a single lipid and how we calculated cluster size distributions based upon the friction factor of an E-cad monomer. We agree that this was unclear in the original submission.

Notably, the diffusion and FRET dwell time analyses do not assume a specific SLB binding valency in any way. In particular, the FRET state of a molecule does not depend upon the valency of the hexahistidine-tag binding to the supported lipid bilayer.

Controls –Many experiments lack sufficient controls to corroborate the conclusions that the authors make:1) The assignment of high-FRET and low-FRET states to clustered and unclustered E-cad, respectively, would be strengthened by accompanying controls of proteins that are known to cluster and known not to cluster. A leucine zipper may be a good control for the high-FRET state.

We thank the reviewer for this suggestion. However, we believe that the FRET measurements (including the existing controls) provide direct evidence of association and dissociation, and we respectfully do not believe that additional experiments with completely different proteins would provide additional support for this conclusion. By imaging control samples without donor E-cad or without acceptor E-cad and comparing the FRET signal to samples with both donor and acceptor E-cad we have demonstrated that the FRET signal we observe is due to binding between donor and acceptor E-cad. The high-FRET state must correspond to the bound or clustered state, which is presumably composed of many microstates that we do not attempt to resolve. Additionally, the low coverage control experiment that we have added to the revised manuscript (subsection “FRET Sample Preparation”) serves as an additional negative control, where extremely short-lived binding is expected. This demonstrates that the FRET signal we observe at high surface coverage is representative of coverage dependent cis-clustering. Moreover, the direct comparison between wild-type and cis-mutant E-cad represents an internal control with results that are as expected.

Moreover, we note that we have previously performed numerous conceptually similar intermolecular FRET analyses to measure binding and association, where the high-FRET state is shown to correspond to the bound state. For example, we have compared binding between complementary and non-complementary DNA using single-molecule FRET (DOIs: https://doi.org/10.1021/acsnano.9b02157, https://doi.org/10.1016/j.jcis.2020.01.070, https://doi.org/10.1021/acs.langmuir.7b02675, https://doi.org/10.1002/anie.201603458, https://doi.org/10.1103/PhysRevLett.116.098303). This work demonstrated that SM-FRET could be used to quantitatively distinguish subtle differences between binding due to specific and non-specific interactions. Similarly, single molecule FRET has been used to infer protein binding (DOIs: https://doi.org/10.1021/acs.biomac.5b00869, https://doi.org/10.1021/bm401302v, https://doi.org/10.1021/jacs.7b03978). To clearly indicate to future readers, the ability to use intermolecular FRET as an indicator of binding, we have added references to these manuscripts in the first paragraph of the Results section.

We would like to emphasize that we do not assign high-FRET and low-FRET states to clustered and unclustered E-cad, as suggested by the reviewer. In fact, we strictly avoid stating that the low-FRET state corresponds to unclustered E-cad, because this is not strictly the case and this assumption is not necessary to measure the dissociation rate, which is the main parameter obtained from the dynamic FRET analysis. While it is true that any unclustered E-cad will be in the low-FRET state, clustered E-cad could also be in the low-FRET state (i.e. donor E-cad bound to unlabeled E-cad). Since we cannot conclusively determine when a donor E-cad molecule is unclustered, we do not attempt to analyze association kinetics. On the other hand, the high-FRET state must correspond to clustered E-cad, as this is the only way we would observe emission in the acceptor channel. Therefore, we can use this information to quantify dissociation kinetics and relative diffusion.

2) The authors claim that monomers diffuse in the supported lipid bilayer at 0.6 μm^2^/s. This is quite slow for His-conjugated proteins. They also claim that monomers are strictly conjugated to a single lipid. This claim would be substantiated if the diffusion of lipids on these bilayers was also measured, but a strictly monomeric conjugation is unlikely given the conjugation chemistry used. Previously, this group has measured a lipid diffusion of 3 μm^2^/s for a similar lipid composition (Cai et al., 2016). The slow diffusion could very well come from the heavily etched glass. Tracking lipids may help the authors reconcile the unexpectedly slow monomer diffusion.

We appreciate the opportunity to clarify this apparent misunderstanding. As described above, we have indeed added new data demonstrating the consistency of E-cad and lipid diffusion.

The value of 0.6 μm^2^/s referred to by the reviewer is the average diffusion coefficient for the mutant and intermediate coverage wild-type conditions in the low-FRET state. However, as described above, we strictly avoid interpreting the low-FRET state as purely monomers, as this is not the case. Some clusters (without acceptor E-cad) will also be in the low-FRET state. This fact is not relevant to any of the main conclusions of the manuscript. The short-time diffusion coefficient in the low-FRET state is faster than in the high-FRET state due to the inclusion of monomers in the low-FRET state and exclusion of monomers in the high-FRET state, but one cannot simply assign that average low-FRET diffusion coefficient to monomeric E-cad. In the presumed presence of a heterogeneous ensemble of diffusing objects, one can estimate the diffusion coefficient of monomers by inspecting the trajectory friction factor distributions we have added as Figure 5—figure supplement 4 using raw data from Thompson et al., 2019. The friction factor peak at ~0.5 s/µm^2^ seen for all E-cad conditions represents monomer diffusion. Therefore, as a rough estimate, monomers diffuse at approximately ~2 µm^2^/s.

In the response above, we explained in detail why the average diffusion coefficient reported is lower than traditionally expected for lipids or supported lipid bilayer bound proteins because of the trajectory filtering procedure that was applied. In particular, it was necessary to apply different criteria to enable the observation of diffusion over more than 2 orders of magnitude. Again, as described above, in the revised manuscript, we have re-analyzed the raw trajectory data using a more traditional filtering criterion, demonstrating that the actual diffusion of lipids and E-cad is similar to that in other reports, and the reported absolute value is simply sensitive to the subtle details of the trajectory analysis. Importantly, we note that this absolute value does not affect any of the findings of this manuscript, which focus instead on comparisons between wild-type and mutant E-cad. Furthermore, the FRET measurements, which provide information about unbinding rates, are unrelated to measurements of diffusion.

Finally, we have added an ensemble-time-averaged MSD analysis (Figure 5—figure supplement 3) using raw data from Thompson et al., 2019. The MSD analysis shows larger diffusion coefficients than the short-time analysis for reasons described in that manuscript. The MSD analysis also indicates E-cad is primarily conjugated to a single lipid across the surface coverage range studied here. The section describing cluster size calculations (Materials and methods) has been rewritten to describe how we determined E-cad are monovalently bound to the supported lipid bilayer and we have removed all instances where we previously stated E-cad was strictly bound to a single lipid, as there likely are a small number of multivalently bound E-cad at all conditions.

3) Related to point 2 above, the authors cite Knight et al., 2010, to justify usage of a linear scaling relationship between mobility and size in the supported membrane. The Knight et al., paper examines protein domains binding to PI lipids and is not directly relatable to the potentially multivalent His-Tag interactions with Ni-NTA lipids. Especially in light of a significantly slower than usual diffusion coefficients the authors see in their experiments, a substantial amount of extra drag or defects appear to be influencing the cadherin motion. Use of constitutive monomer dimer controls would really help solidify this aspect of the work e.g. as in Chung et al., Biophys. J. 2018 [doi.org/10.1016/j.bpj.2017.10.042].

As described above, we have shown that E-cad is primarily bound to a single lipid via MSD analysis included here and displacement distribution analysis that was published previously (Thompson et al., 2019). Therefore, the additive friction factor model is expected to provide a good approximation for cluster sizes. We have explained above why the average absolute short-time diffusion coefficients are lower than expected and why we do not simply remove slowly diffusing molecules.

A critical point is that we have designed our analysis to enable observation of a large dynamic range of cluster sizes (which is not the typical goal of SM tracking experiments for bilayer associated proteins) and are not generally focused on distinguishing between monomers, dimers, etc. using diffusion. Using diffusion to distinguish small order oligomers is difficult when the interactions are dynamic. The primary purpose of the cluster size distributions is to indicate the range over which we see clustering, and provide a means of comparison to the Monte Carlo simulations. We have added clarification on how the reader should interpret the cluster size distributions in the Results section.

4) Figure 3: The rates of photobleaching and desorption are not measured, and so it is unclear if they occur on the same time scale as dissociation. The authors state that the curve in the plot represents multiple modes of dissociation, but they could also represent a convolution of photobleaching, desorption, and dissociation on similar time scales. The Materials and methods section states that a 50 mW diode-pumped laser was used as the illumination source, but what was the illumination power at the sample? The rates of photobleaching and desorption should be included as a supplement to this figure and it should be discussed how, if at all, these processes complicate data analysis.

This is an important point and as the reviewer mentions, an analysis of the trajectory observation times (limited by photobleaching and desorption) is necessary prior to quantitative interpretation of the high-FRET state dwell times previously shown in Figure 3A. In fact, because such an analysis is so complex and requires various assumptions, we did not use the dwell time distributions to estimate unbinding rates, and instead employed the three-state Markov model, which we have previously found to be more reliable for the determination of apparent folding and unfolding rates in intramolecular SM-FRET experiments of adsorbed and immobilized FRET-labeled proteins (DOIs: https://doi.org/10.1021/acsnano.8b02956, https://doi.org/10.1021/jacs.9b11707).

We had originally thought to include the dwell-time distributions because they serve to demonstrate the qualitative trends and may be more intuitively accessible for some readers than the Markov analysis. However, based upon the points mentioned by the reviewers, we feel that explaining all of the complications and limitations associated with interpreting these distributions would be too complex and distracting from the main messages of the manuscript, so we have decided to dramatically reduce the discussion of the high-FRET dwell time distributions and put the focus where it belongs … on the parameters obtained from the Markov analysis. We have therefore removed Figure 3A showing the high-FRET dwell time distributions, although they are still included in Figure 3—figure supplement 1 for qualitative comparison to each other and the low coverage control dwell times. Additionally, the complicated interpretation of the curvature of the dwell time distributions has been removed from the Results subsection “Nonspecific Cis-Interactions Dissociate Faster than Specific Cis-Interactions” and here we explain that it is difficult to extract quantitative information from the dwell time distributions for reasons the reviewers suggest, among others, which is why a Markov model is used to quantitatively analyze the dissociation kinetics.

The interpretations previously made based upon the dwell time distributions actually did not provide unique information. The diffusion analysis still indicated the presence nonspecific interactions, and the average dissociation rate constants determined by using a three-state Markov model (which are not determined using the dwell time distributions) still indicate that the high-coverage wild-type conditions exhibits slow dissociation.

5) WT and mutant E-cad are labeled in a manner that does not control the stoichiometry of the label to protein. The WT and mutant E-cad labeled with 647 have notably different labeling efficiencies (2.3 and 1.3, respectively). How do these differences in labeling efficiencies affect the interpretation of the dwell time distributions, if at all?

This is a good point and as mentioned directly above, because of complications like this, we have removed all of the quantitative conclusions based upon the dwell time distributions. Differences in acceptor labeling efficiency very well may affect the dwell time distributions.

6) Related, the donor WT E-cad-AF555 has a labeling efficiency of 1.3. Single E-cad molecules therefore could have 1, 2, or 3 fluorophores. Can the authors distinguish between the dwell of a single donor E-cad molecule with two fluorophores and two diffraction-limited donor E-cad molecules, each with one fluorophore? It seems like with their current labeling scheme, the authors would expect to be sampling multiple populations of dwelling species, even if photobleaching and desorption were corrected for. Because of these complications in how the experiment was conducted, we do not find strong evidence in Figure 3A that cis-interactions exhibit slow dissociation.

As mentioned above, we have removed essentially all conclusions based upon the dwell time distributions, as there are several complications in extracting quantitative information from these distributions.

Importantly, the average dissociation rate constants determined via the three-state Markov modelling account for trajectory observation times (i.e. trajectories ending due to desorption or photobleaching), and any differences in observation times between configurations, and therefore are expected to provide accurate rates (see Materials and methods subsection “FRET-State Dwell Time Distributions and Transition Rate Determination”). Therefore, it is still apparent that the high-coverage wild-type condition exhibits slow dissociation, relative to the mutant and intermediate surface coverage wild-type conditions. We have rewritten this subsection of the Results (“Nonspecific Cis-Interactions Dissociate Faster than Specific Cis-Interactions”) to avoid making quantitative claims based purely on the dwell time distributions, and we have added additional clarification that the rate constants (Figure 3) are not calculated using the dwell time distributions.

7) The authors seem to use diffusion coefficient as the only measurement of E-cad clustering size. Could these measurements be corroborated by a separate measurement of cluster size, say acceptor fluorescence intensity? For example Chung et al., Nature 2010 [doi.org/10.1038/nature08827] used a simple linear scaling assumption for EGFR diffusion as a function of cluster size to claim dimer was the primary species, whereas later single molecule photobleach analysis revealed they were largely higher order oligomers, not well distinguished by mobility (Huang et al., eLife 2016 [doi: 10.7554/eLife.14107]). Mobility alone, especially in a 2D membrane environment, can be unreliable.

Previously we used a number of parameters to indirectly observe a trend of increased clustering, such as: the short-time diffusion coefficient, the time-averaged diffusion coefficient, the anomalous diffusion coefficient, and the average surface residence time (DOI: https://doi.org/10.1021/acs.jpclett.9b01500). However, these parameters do not allow direct measurement of individual cluster sizes. We use mobility as a measure of cluster sizes as it is an extremely good measure of individual cluster sizes over a very large dynamic range, which is what we are most interested in here. Single molecule photobleaching analysis may be a more reliable method to distinguish between small order oligomers, however using photobleaching to determine the cluster size of a cluster of ~100 molecules is not practical for a dynamic system. Thus, in the two papers referenced by the reviewer, photobleaching analysis may be a more reliable method to distinguish between the dimers and small oligomers, but in this work, we observe an increasing number of extremely slow diffusing objects corresponding to very large clusters. Our focus is not necessarily to differentiate between small order oligomers using this model, but to instead approximately determine the entire distribution of cluster sizes over a wide dynamic range and we believe that mobility is the best approach for this.

Alternatively, using acceptor fluorescence intensity to infer cluster sizes, as the reviewer suggests, is not plausible using this experimental system as the vast majority of E-cad present within a cluster are not labeled with a fluorescent tag and therefore only contribute to the mobility of the cluster and not the fluorescence intensity in either channel. The use of unlabeled E-cad is necessary due to investigate high coverage conditions, while keeping background emission at a minimum. We have added clarification on why we used mobility to infer cluster sizes in the section describing cluster size calculations to the Materials and methods section.

Imaging Experiments –1) The imaging buffer isn't directly specified. It is left for readers to assume that the imaging buffer is the same as the HEPES buffer that the protein was labeled in. Is this the case? If so, then the lack of an oxygen scavenging system in the buffer may cause complications, such as photo-induced protein crosslinking (Chung et al., 2016). Do the step size distribution and dwell time distribution results change between the 1st and 10th acquisition on the same bilayer? Do results vary with varying laser power?

We thank the reviewer for noticing that we failed to explicitly describe the imaging buffer. A description of the imaging buffer has been added in the Materials and methods subsection describing “FRET Sample Preparation”. An oxygen scavenger system was not used. We have tested oxygen scavenging systems in the past, but they have not provided significant benefit at the typical laser powers and low concentrations of labeled species we use. Nevertheless, we have tested for any significant trends in both the step size distributions and the dwell time distributions with imaging time (i.e. video number) to test for complications due to the lack of an oxygen scavenging system. We have included both the single video step size and dwell time distributions as Figure 1—figure supplements 2-3, respectively. Neither step size nor dwell time distributions exhibited a systematic trend in general distribution behavior with increasing imaging time. This observation suggested that the lack of an oxygen scavenger system in the imaging buffer did not cause significant issues. A reference to these additional results has been added to this paragraph of the Materials and methods section where the imaging buffer is described (subsection “FRET Sample Preparation”).

Result dependence upon laser power was not tested and we believe the distributions as a function of imaging time indicate that photo-induced complications are not present for this system. Also, based upon the extremely low fraction of donor labeled E-cad on the bilayer (~2x10^-6^) we would expect photosensitization-based complications to be insignificant, as suggested by Chung et al., 2016.

2) The calcium content of the imaging buffer should also be explicitly noted, as it is important in interpreting the measured diffusion coefficients.

We have explicitly stated the imaging buffer concentrations used for all conditions in the Materials and methods subsection titled “FRET Sample Preparation”. We also make note of the high calcium conditions here.

3) The donor is tracked for single molecule tracking and classification of high-FRET and low-FRET states. What is the E-cad-AF555 density for the "low densities" used in single particle tracking experiments?

The donor (E-cad-AF555) surface density was ~0.003 E-cad/µm^2^ for all three experimental conditions. We have added a sentence stating this value in the Materials and methods subsection title “FRET Sample Preparation”.

4) In their Materials and methods section, the authors claim that "due to a combination of bright contaminants and inherent defects in the supported lipid bilayers, a permanently immobile (or highly confined) population was observed in the donor channel." Do the authors know if they have mobile fluorescent contamination? This could be assessed by taking a video with the same acquisition parameters used for step size distribution analysis before adding fluorescent E-cad.

We did image samples free of fluorescent E-cad to observe any fluorescent contamination present in the supported lipid bilayers (subsection “FRET Sample Preparation”). The contaminants observed were primarily immobile. However a small number were mobile. Noticeably, the contaminants were generally visible only in the donor channel and were inherently brighter than the population of interest (individual E-cad with a single AF555 label). As we did not want to directly remove trajectories based on mobility, we choose to remove the majority of contaminants and anomalous trajectories using a median intensity cutoff. Furthermore, the contaminants were observed to quickly dissociate from the supported lipid bilayer, relative to the more strongly bound Ni-His6 bound E-cad. Therefore, we choose to use an extended surface residence time (trajectory observation time) filtering criterion as well as the intensity cutoff.

It is clear in retrospect that the paragraph on bilayer contamination and trajectory filtering did not clearly describe what kinds of anomalous trajectories were observed and how they were removed from the data analysis. We have rewritten this paragraph (subsection “Image Analysis”) to better describe the trajectories we wished to remove, why we needed to remove them, and how we went about filtering out these anomalous trajectories.

5) Related to 6 (below), could the contaminants be contributing to the FRET signal?

The contaminants did not significantly contribute to the FRET signal as nearly all of the contaminants were solely in the donor channel, not in the acceptor channel (subsection “Image Analysis”). Furthermore, direct excitation of acceptor labeled E-cad was also an insignificant contribution to the FRET signal. This was verified via imaging a sample without donor labeled E-cad, but including unlabeled and acceptor labeled E-cad (subsection “FRET Sample Preparation”). Lastly, the low coverage control configuration that we have added to the manuscript shows that we are in fact observing surface coverage-dependent interactions via FRET (subsection “FRET Sample Preparation”).

6) FRET analysis is complicated by donor or acceptor molecules that have more than one fluorophore. What steps are taken to filter proteins with more than one dye out of analysis (Hanson, J.A., and Yang, H, J. Phys. Chem. B, 2008 [oi.org/10.1021/jp804440y])?

Based on the analysis approach used in this manuscript, it is not necessary to remove proteins with more than one dye in order to observe transitions from high to low FRET states.

The manuscript the reviewer suggests discusses intramolecular FRET trajectories in which a molecule is labeled with both donor and acceptor fluorescent labels. Such an intramolecular FRET system would indeed be complicated by nonspecifically labeling proteins with more than one donor or acceptor dye. However, this is not the case for this manuscript, because we are using intermolecular FRET and E-cad molecules that are labeled with either donor or acceptor fluorophores, or are not labeled at all. Furthermore, as described above, we do not use the quantitative FRET efficiency to calculate quantitative distances, but only to infer when a donor E-cad molecule is associated to an acceptor E-cad molecule.

To clarify this analysis, in the revised manuscript we have added FRET “heat maps” of acceptor and donor intensities to Figure 1A and Figure 1—figure supplement 1. These heat maps use all molecular observations for the three experimental conditions studied. They exhibit two peaks, indicating two readily distinguishable populations: a low-FRET population at high donor intensity and low acceptor intensity, and a high-FRET population at high acceptor intensity and low donor intensity. A simple algorithm is employed to divide these FRET maps into high-FRET and low-FRET regions, as indicated by the dividing lines in the figures. Most importantly, the high-FRET population represents acceptor E-cad associated with a donor E-cad, regardless of whether more than one dye is attached to a given molecule. The transition from high-FRET to low-FRET involves a dissociation event. The low-FRET population may be composed of a combination of both associated (i.e. donors bound to unlabeled E-cad) and unassociated donor E-cad.

Also, the low-FRET state is centered around an acceptor intensity of zero and the high-FRET state is centered around a donor intensity of zero. This implies that the two populations approximately correspond to either complete energy transfer (high-FRET) or negligible energy transfer (low-FRET). We do not attempt to further separate the high-FRET population into the many microstates that are likely present (dimer, trimer, etc.), but instead quantify the distribution of dissociation rates representing transitions from the high-FRET population to the low-FRET population (i.e. dissociation of the donor from the acceptor). Therefore, due to the presence of two discrete populations corresponding to either complete energy transfer, or no energy transfer, including donor or acceptor E-cad with multiple fluorescent labels will not complicate this analysis when accounting for trajectories ending due to photobleaching or desorption.

We employ a median donor intensity exclusion criterion to exclude trajectories with a median donor intensity above the 60^th^ percentile. This exclusion criterion was selected for a number of reasons: fluorescent contaminants were observed to be inherently brighter than single E-cad and removal of aggregates of multiple donor E-cad to ensure the tracking of single-molecules. However, this exclusion criterion also will remove donor E-cad with multiple fluorophores. Overall, this intensity cutoff ensures that we are only analyzing single E-cad that have been labeled with one AF555 donor fluorophore. A distribution of the median donor intensities has been added to the manuscript as Figure 1—figure supplement 7 showing the 60^th^ percentile cutoff value for all conditions. The peak at high donor intensity must correspond to single donor-labeled E-cad with only one label and the cutoff removes trajectories corresponding to bright contaminants, E-cad donor aggregates, and E-cad with multiple fluorescent labels. A similar intensity exclusion criterion is frequently used to remove donor aggregates and anomalous trajectories in single-molecule tracking experiments (DOIs: https://doi.org/10.1016/j.bpj.2010.08.046, https://doi.org/10.1016/j.bpj.2008.10.020).

Image Analysis –All of the image analysis conducted in this work was built in-house by this research team. While this alone is not an issue, we are concerned the algorithms have not, to our knowledge, been verified with simulations for which the ground truth is known, nor have they been compared to established, verified, and widely-used particle localization and tracking algorithms (Serge A., et al., Nat. Methods 2008 [doi: 10.1038/nmeth.1233]; Jaqaman, K., et al., Nat. Methods 2008 [doi.org/10.1038/nmeth.1237]; Chenouard, N., et al., Nat. Methods 2014 [doi.org/10.1038/nmeth.2808]; Tinevez, J.-Y.,et al., Methods, 2017 [doi.org/10.1016/j.ymeth.2016.09.016]). We have the following questions about the authors' particle localization and tracking algorithms:

In general, we would like to point out that although the software used for all image analysis has been written in-house, the underlying methods and algorithms for tracking and localization are well-established and have been used extensively. For a detailed description of the automated object thresholding algorithm, see Kienle and Schwartz, 2019. For details on the tracking algorithm, see Marruecos, D.F, et al., ACS Macro Lett. (DOI: https://doi.org/10.1021/acsmacrolett.8b00004) and Kienle et al., 2018. We have developed our own tracking software to streamline analysis, allow for customization where needed, and permit rapid, integrated analyses of large data sets (often including 10^5^ trajectories), but the mathematical algorithms used are not themselves novel. Furthermore, our tracking software allows for the integration of FRET and tracking, which is not possible using standard established tracking packages such as those suggested by the reviewer. Over the years, we have cross-validated our software with several software packages that analyze FRET and positional tracking separately, but to date, no other software packages satisfy all of our needs. Moreover, as algorithms are added and modified over the years, we routinely validate them using simulated data. We have recently provided a free license to our software to another research group, and now that the license document has been prepared and approved by our university, we are happy to consider extending this courtesy to other groups, with the understanding that we are not able to provide technical support beyond the documentation included in the code.

In the first paragraph of the “Image Analysis” section we have included references to the prior works describing the tracking methods used, and a statement informing the reader that the methods and algorithms used for tracking and localization are established and were written into our in-house software to greatly improve the analysis throughput and accessibility.

Using in-house software that integrates tracking with single-molecule FRET is extremely beneficial and allows for a diverse number of applications. For a small number of recent representative examples of how we have applied this software, please see Sarfati, R., Schwartz, D.K., ACS Nano 2020 (DOI: https://doi.org/10.1021/acsnano.9b07910), Weltz, J.S., et al., J. Am. Chem. Soc. 2020 (DOI: https://doi.org/10.1021/jacs.9b11707), Traeger, Lamberty and Schwartz, 2019, Weltz, J.S., et al., ACS Catal., 2019 (DOI: https://doi.org/10.1021/acscatal.9b01176).

1) Do they check for trajectories in which two donors overlap and filter their data to exclude those events from their dwell time distributions or otherwise handle this merging and splitting behavior?

For this particular analysis, merging and splitting behavior is ignored because these instances are indistinguishable from bleaching/desorption events and bleaching/desorption events are much more likely to occur. To quantitatively justify this, we have estimated our merging error rate based upon the tracking radius of 3 pixels, the object density of ~0.003 molecules/µm^2^, and a bleach/desorption probability of 0.3 estimated from trajectory observation time distributions. Based upon these experimental values, only 0.2% of observations are expected to have two objects within 2-times the tracking radius of each other. The merging probability for these two objects is 7%, and the probability of one of the two bleaching is 21%. Therefore, only 0.07% or 1/1400 observations comprise merging events and could prematurely truncate a trajectory.

2) Have they quantified their tracking error rate using simulated data for which the ground truth is known? If so, how closely do the simulated data reflect the actual data that the authors have acquired (e.g. in diffusion rate, bilayer density, type of motion)?

As mentioned above, the localization and tracking algorithms used for this study are standard. Please see Marruecos, D.F, et al., ACS Macro Lett. (DOI: https://doi.org/10.1021/acsmacrolett.8b00004) for detailed discussion of object localization and the trajectory linking algorithm. See Kienle et al., 2018, for details on intensity uncertainty estimation from camera shot noise. Lastly, see Kienle and Schwartz, 2019, for details on the automated object thresholding algorithm. As new algorithms are integrated into our tracking code, they are indeed validated using simulated data as a standard part of the debugging process.

3) Have other groups used this algorithm? If so, it would be useful for the authors to cite these papers.

No other groups have published with our software package to date.

4) The three pixel tracking radius that the algorithm allows seems huge given the pixel size and diffusion coefficients of the species. This large tracking radius could cause complications depending on the density of the donor E-cad. Why was this radius chosen and does it result in tracking artifacts?

The three pixel tracking radius was selected to allow for tracking of rare, large displacements corresponding to the tail of the step-size distributions, without being so large that it results in tracking artifacts. At the low donor E-cad surface density of ~0.003 E-cad/µm^2^, a tracking radius of 3 pixels will not result in a significant number of artifacts, but it allows observations of large displacements as indicated by Figure 2A-C, Figure 1B, H, and Figure 2—figure supplement 1. As described above, based on the experimental object density and tracking radius, only 0.2% of observations are expected to have two objects within 2-times the tracking radius of each other. Therefore, the 3 pixel tracking radius does not result in significant tracking artifacts for the experimental object density, but does allow for the observation of diffusive steps over a large dynamic range, as is necessary.

5) The 2 pixel, or approximately 800 nm, distance requirement to identify FRET pairs also seems far too big. 800 nm is far out of FRET range of 1-10 nm. We understand that there is some uncertainty in localization, but even so, an 800 nm threshold does not make sense. Papers identifying co-locomotion of two associated membrane proteins often use a much smaller (100 nm) co-localization threshold (Wilkes, S., et al., Science, 2019).

We thank the reviewer for pointing out that justification of the 2-pixel colocalization radius is needed. A radius of 2 pixels was selected to allow for potential colocalization of objects with a large position uncertainty, while also allowing for potential channel registration error. However, colocalization is only used for intermediate FRET when objects are observed within the 2-pixel radius in both donor and acceptor channels (i.e. we are not looking at co-locomotion). If an acceptor object is observed, it must be associated with a donor molecule, so co-locomotion is not necessary.

We have added a representative histogram of observation position uncertainty as Figure 1—figure supplement 6, indicating that a number of observations have a position uncertainty around 1 pixel. Therefore, the colocalization distance was set to 2 pixels to allow for colocalization of observations with high uncertainty. Based upon the relatively fast diffusion and low intensity of the objects of interest, it is expected that a number of observations will have a relatively large position uncertainty. Furthermore, the FRET maps (Figure 1—figure supplement 1) show two population peaks, both centered around either zero donor intensity or zero acceptor intensity, indicating that colocalization is rare and molecules typically either essentially exhibit complete energy transfer or zero energy transfer (not necessarily surprising for intermolecular FRET of bound molecules). If the colocalization distance of 2 pixels was too large, resulting in erroneous FRET pair assignment, one would expect to see significant peaks centered around both high donor and acceptor intensities. Also, considering the extremely low density of fluorescent objects in both channels, it is not expected that a 2-pixel colocalization radius would result in erroneous FRET pair assignment as significantly less than 0.2% of observations are expected to have both donor and acceptor objects within 2 pixels. We have added justification for the selection of a 2-pixel colocalization radius in the first paragraph of the Materials and methods subsection titled “Image Analysis” for future readers. Also, we believe that the raw videos will indicate that this parameter value is not an issue.

6) Do the authors require that the donor and acceptor signal co-locomote when measuring association times? For how many frames and what distance needs to be maintained to classify the signals as a FRET pair?

As discussed above, intermediate FRET where both donor and acceptor emissions are above background are extremely rare (Figure 1—figure supplement 1) for this system, and besides those situations, we do not look for colocalization. However, in the instance of intermediate FRET where donor and acceptor signals are colocalized within the 2-pixel radius, they must remain within this colocalization radius to be considered a FRET pair. For the vast majority of molecular observations in either donor or acceptor channels, there is not an object in the opposite channel within the colocalization radius. This indicates either essentially complete energy transfer or zero energy transfer and the missing intensity (either donor or acceptor) is taken as the background intensity of the corresponding pixels in the opposite channel.

The association times were calculated as the time periods a trajectory was in the high-FRET state as a trajectory will only be in the high-FRET state if a donor is associated with an acceptor (i.e. direct excitation of acceptor E-cad was not used). All observations of all trajectories were assigned to either the high-FRET (associated) or low-FRET state based on the FRET map and the dividing line separating the two peaks. A FRET state change was only assigned if both donor and acceptor intensity crossed the dividing line by more than the uncertainty of corresponding intensity. As referenced in the first paragraph of the “Image Analysis” section, please see Chaparro Sosa et al., 2018, as another example and for addition details on FRET state analysis. To improve understanding of these methods in the revised manuscript, we have added a representative FRET map used for state assignment as Figure 1A (additional FRET maps added as Figure 1—figure supplement 1), and a concise description of the two FRET populations observed was added to the first paragraph of the Results section.

7) The step size distribution is built from localizations in adjacent frames. However, depending on the localization algorithm, artifacts can be introduced in a step size distribution if the precision of particle localization is large compared to the size of steps taken. (Cohen, E.A.K., et al. Nat. Comm. 2019 [DOI: 10.1038/s41467-019-08689-x]; Hansen, A.S., eLife, 2018 [doi: 10.7554/eLife.33125]). We recognize that the authors are most interested in short-time scale diffusion, but the authors can test how accurate the adjacent frame step size distribution is by looking at the distributions from step sizes calculated from every other or every third frame.

Importantly, we would like to again point out that our localization and tracking algorithms are established methods and are not unique to our implementation. We have previously inspected displacement distributions as a function of lag times for E-cad diffusion on supported lipid bilayers (DOI: https://doi.org/10.1021/acs.jpclett.9b01500, Figure 1—figure supplement 6). The distributions exhibit a central peak with heavy tails. The central peak was determined to represent displacements on the order of the position uncertainty as the central peak did not broaden with increasing lag time. Similar dynamics are expected for the system studied here, where a number of small displacements corresponding to immobile periods or large cluster diffusion are on the order of the position uncertainty and large displacements corresponding to monomer or oligomer diffusion are greater than the position uncertainty. One can visually observe this behavior by inspecting step size distributions (Figure 2A-C and Figure 2—figure supplement 1) and a histogram of observation position uncertainty (Figure 1—figure supplement 6) in tandem. Generally, the observations with high position uncertainty are due to motion blur from fast diffusing trajectories.

Data analysis –1) The step size distribution is fit with three diffusive states. Why are three states chosen and do the authors have a physical interpretation of what those states are?

Three diffusive states were used to fit all complementary cumulative squared displacement distributions as this was the minimum number of diffusive states that resulted in randomly distributed residuals. However, a three term Gaussian mixture model was primarily selected because of the ability of such a model to serve as a robust fitting function to extract an accurate average short-time diffusion coefficient in all cases. One could interpret such a model as a superposition of Brownian diffusive modes, but we strictly avoid such an interpretation here and are purely interested in the average short-time diffusion coefficient. We have added clarification on why we selected a Gaussian mixture model with three terms in the Materials and methods section describing average short-time diffusion coefficient determination (subsection “Average Short-Time Diffusion Coefficient Determination).

2) Can the authors justify why they are interested in the short time diffusion coefficient, D¯short, as opposed to the overall diffusion coefficient, D? Can they provide citations for how this method has been used previously to put their analysis in context?

We are interested in the short-time diffusion coefficient as this parameter represents the average instantaneous molecular diffusion coefficient on the shortest experimentally accessible time scale. This is especially of interest as opposed to the overall diffusion coefficient because of the ability to categorize single-frame displacements into either FRET state. The lateral associations we observe in this work are dynamic and frequently show transitions within a single trajectory. Therefore, using another parameter such as the overall diffusion coefficient would not allow us to breakup trajectories with transitions between FRET states. For examples on how this short-time diffusion coefficient analysis has been used previously, see https://doi.org/10.1021/acsami.8b05523, https://doi.org/10.1021/acs.biomac.5b00869, and https://doi.org/10.1002/admi.202000533. We have added a sentence in the subsection “Nonspecific and Specific Cis-Interactions Are Present in E-cad Clusters” stating why the short-time diffusion coefficient is of most interest for this system and included references to manuscripts where this method has been used previously.

3) In the “Image Analysis” subsection of the Materials and methods section, the authors state that they analyze particle trajectories with median donor intensities in the bottom 60%. Why was the 60% cutoff chosen? Approximately how many fluorophores does this intensity threshold correspond to? What is the probability that they are tracking two closely associated E-cad molecules and not a single molecule?

The median donor intensity 60^th^ percentile exclusion criterion was selected for a number of reasons: fluorescent contaminants were observed to generally be inherently brighter than single E-cad and removal of aggregates of multiple donor E-cad to ensure the tracking of single-molecules. However, this exclusion criterion also will remove donor E-cad labeled with multiple fluorophores. Overall, this intensity cutoff ensures that we are primarily analyzing single E-cad that have been labeled with one AF555 fluorophore. A distribution of the median donor intensity has been added to the manuscript as Figure 1—figure supplement 7 showing the 60^th^ percentile cutoff value for all conditions. The large peak at high donor intensity must correspond to single donor-labeled E-cad with only one label because the labeling efficiency of ~1.3, and the cutoff removes trajectories corresponding to bright contaminants, E-cad donor aggregates, and E-cad with multiple fluorescent labels. The paragraph describing the trajectory filtering procedure (subsection “Image Analysis”) had been rewritten to better indicate why the trajectory exclusion criteria were selected.

4) To substantiate their findings the authors may further discuss certain parameter selections as well as their results in the context of current literature and physiological relevance. For example:a) What is a physiologically relevant surface coverage that is not within cell-cell adhesion? The authors mention ~49,000 E-cad/μm^2^ but within cell-cell adhesion the surface coverage is expected to be significantly higher because of diffusion trap. However, the experiment setting in this paper tests cis interactions independently of trans interaction.

The average surface coverage of E-cad expressed on MDCK cells has been estimated to be ~17 E-cad/µm^2^ via quantitative flow cytometry (DOI: 10.1242/jcs.105775), although we would suggest the surface coverage within junctions is the more relevant value. It is certainly the case that E-cad surface coverage is greatly elevated within cell-cell adhesions due to diffusion trapping, however it is expected that cis-interactions facilitate the drastic accumulation in the adhesive zone due to lateral binding to trans-dimers; i.e. without cis-interactions junction accumulation would be much more subtle (DOI: https://doi.org/10.1073/pnas.1011247107). More importantly, the distribution of E-cad within cell-cell junctions is highly heterogeneous and consists of a distribution of surface densities, as well as both adhesive and non-adhesive clusters (DOI: https://doi.org/10.1016/j.devcel.2014.12.003). Where, the non-adhesive clusters are stabilized via cis-interactions largely independent of trans-interactions, but still experience the locally high surface coverage environment within the junction. Therefore, we would argue that quantifying cis-interactions independent of trans-interactions at a surface coverage relevant to cell-cell junctions is directly related to the dynamics, stability, and kinetics of cell-cell adhesions. We have added additional clarification on why the surface coverage range we study is physiologically relevant in the second paragraph of the Results section in a concise manner.

b) We were surprised by the large size of clusters formed without trans interactions. Previous studies have reported much smaller micro clusters – of ~5 molecules if any (Zaidel-Bar lab). Could the authors discuss the cluster sizes observed in their experimental setting.

This is a good point. It is important to put the cluster sizes we observe into context. We have added additional discussion of the large clusters observed in the second paragraph of the Discussion section. Importantly, the study referenced (DOI: https://doi.org/10.1016/j.devcel.2014.12.003) does in fact observe the formation of clusters of a comparable size, independent of trans-interactions (~30 – ~50 E-cad). However, the median of the cluster size distributions they observe differs from the cluster size distributions we report here. There are a number of differences between their in vivo system and our in vitro model that could explain this, such as membrane mobility, E-cad surface coverage, and/or the dynamic range of cluster size determination techniques.

Hidden Markov Modeling –1) Work by J. Elf is relevant to the Hidden Markov Modeling performed in this study and should be cited (Persson, J., et al. Nat. Methods, 2013 [DOI: 10.1038/nmeth.2367]).

We thank the reviewer for suggesting this manuscript for reference, however, we do not use a hidden Markov model. The state of the trajectory is not inferred based on the diffusion of the trajectory, as is done in the suggested manuscript, but is directly assigned to each observation via FRET map. We have added Figure 1A and Figure 1—figure supplement 1 to illustrate the presence of two populations and show the dividing line used to assign each observation to either the low-FRET population or the high-FRET (associated) population (i.e. the trajectory state sequence is defined prior to modeling). We then use a three state Markov model to extract dissociation rates (state transition probabilities) while accounting for incomplete high-FRET state dwell times at the beginning or end of a trajectory by including a third state corresponding to the end of a trajectory due to desorption or photobleaching. Additional clarification was added on observation state assignment was added to the first paragraph of the Results section. Modeling using a three state Markov model is also discussed in detail in the Materials and methods subsection “FRET-State Dwell Time Distributions and Transition Rate Determination”.

Figures –1) Can the authors justify why they chose to plot average dissociation rate constants in Figure 3B? It would be useful to know what rates they extract for the quick "non-specific" interactions and the longer "specific" interactions for each distribution. Based on the text, it seems like they expect this non-specific interaction to be similar for similar bilayer densities of WT and mut E-cad and that the specific interaction to be similar for all bilayer densities of WT E-cad. Is this the case?

As explained above, the single-molecule FRET association time (high-FRET state dwell time) distributions were not used to determine the average dissociation rate constants, for reasons described in detail in a response to a previous reviewer. These rates were extracted using the Markov modeling. In the original submission, these distributions were used to qualitatively compare interaction lifetimes across the different systems. As mentioned above, the association times used to construct the distributions are not necessarily bounded by dissociation events (e.g., association times observed at the beginning or end of a trajectory). Therefore, this did not allow direct fitting of the distributions previously shown in Figure 3A to say an exponential mixture model as these association times may be convoluted with the surface residence times (desorption and/or photobleaching), which is mentioned by the reviewer above. Additionally, other complications, such as differences in acceptor labeling efficiency and the median donor intensity exclusion criterion, make extracting quantitative information from the dwell time distributions difficult. Upon reflection, we feel that these distributions may confuse a reader more than help them, so they have been moved from the main text to the supplementary information and are only described briefly, putting the correct emphasis on the Markov model.

The three-state Markov model was used to overcome the complexities associated with extracting rates from the dwell time distributions. The incorporation of three states (low-FRET, high-FRET or associated, and off) allowed quantitative calculation of the transition probability from the high-FRET state to the low-FRET state. This transition must correspond to the dissociation of donor and acceptor E-cad and the transition probability is directly related to the dissociation rate constant. Furthermore, the off state represents the end of a trajectory due to photobleaching, desorption from the bilayer, etc. and accounts for these phenomena. The probability of a trajectory ending is independently estimated from measured surface residence time (trajectory observation time) distributions.

Since our single-molecule FRET data only allows classification of observations into the two FRET states, as discussed above, we do not attempt to differentiate between specific and nonspecific interactions and other microstates present within the high-FRET population from the trajectories. The maximum likelihood estimation method assumes a distribution of transition probabilities (Figure 3—figure supplement 4 and Figure 3—figure supplement 6) to account for interaction heterogeneity. Overall, the single-molecule FRET data and maximum likelihood estimation method cannot independently determine the dissociation rate constants for specific and nonspecific interactions, only the average dissociation rate constant for all interactions which is calculated from the distributions shown in Figure 3—figure supplement 4. This is why the kMC simulations are employed, as they are capable of distinguishing specific and nonspecific interactions and can define the on and off rate constants for specific and nonspecific interactions independently. The resulting on and off rate constants for both specific and nonspecific interactions as determined by kMC simulations are presented in the Results subsection titled “Heterogeneous kMC Simulations Differentiate Specific and Nonspecific Interactions”. We have added additional clarification of Markov modeling and average dissociation rate constant calculation to the Results section where the model is introduced in the subsection “Nonspecific Cis-Interactions Dissociate Faster than Specific Cis-Interactions”.